# GRAFT: INTEGRATING THE DOMAIN KNOWLEDGE VIA EFFICIENT PARAMETER SYNERGY FOR LLMS

## ABSTRACT

Large Language Models (LLMs) have achieved success across various domains. However, their applicability tends to degrade when confronted with different types of data inputs, especially for LLMs that have been fine-tuned for specific tasks. Despite its importance, the study of knowledge sharing among domain-specific LLMs—such as those trained for mathematics or code—remains largely underexplored. To address the fragmentation of knowledge across domain-specialized LLMs, we propose a unified parameter integration framework that enables modular composition of expert capabilities. Our method is grounded in a novel Compatibility-Aware Parameter Splicing (CAPS) strategy, which leverages both local functional attribution and global information-theoretic signals to guide selective parameter fusion. By extending this mechanism to the low-rank adaptation layer granularity, we ensure efficient integration with minimal inference overhead. Furthermore, we introduce a domain compatibility scoring mechanism that quantifies inter-expert alignment at the activation level and correlates with downstream task utility. This principled fusion protocol allows the final model to synergize heterogeneous expertise while preserving structural modularity. Extensive evaluations across diverse language and reasoning benchmarks validate the effectiveness of our framework, offering a scalable path toward compositional, domain-adaptive LLMs. Our project is available at `https://anonymous.4open.science/r/Graft-8213`.

## 1 INTRODUCTION

Large Language Models (LLMs) (Liu et al., 2023a; Wang et al., 2024b; Liang et al., 2024) have emerged as a powerful paradigm, demonstrated remarkable success across a wide range of tasks, such as general reasoning, mathematics, programming, and scientific applications(Dyer & Gur-Ari, 2022; Lin et al., 2025; Liu et al., 2025; Hui et al., 2024; Tang et al., 2025). However, most of them cannot excel in all domains, mainly because they are trained on domain-specific settings. Of course, we can introduce more data from different domains and train a comprehensive model from scratch, but it requires significant computational resources. Consequently, there has emerged a recent trend in the research community, *i.e.*, Model Merging(Yang et al., 2024b; Akiba et al., 2025; Li et al., 2023b), focused on exploring methodologies for effectively merging multiple independently trained models without relying on their training data. The practice of model merging has emerged as a promising solution to enhance model generalization.

Broadly, the existing model merging methods rely on direct integration of model parameters(Gupta et al., 2020; Wortsman et al., 2022; Lv et al., 2025), but these methods presuppose uniform architectures across models and often fail to capture the strengths of diverse specialized models. More advanced heuristics like Task Arithmetic(Ilharco et al., 2022) and TIES-Merging(Yadav et al., 2023) fuse parameters in an element-wise fashion, but still fail to adequately address parameter interference or to align heterogeneous representations. These shortcomings are further exacerbated when merging LoRA-tuned models across disparate domains, because the misaligned parameter subspaces and an inability to identify which adaptations are complementary versus conflicting often result in severe performance degradation. Collectively, these limitations highlight the need for a principled model fusion strategy capable of adaptively aligning and integrating multi-domain knowledge.

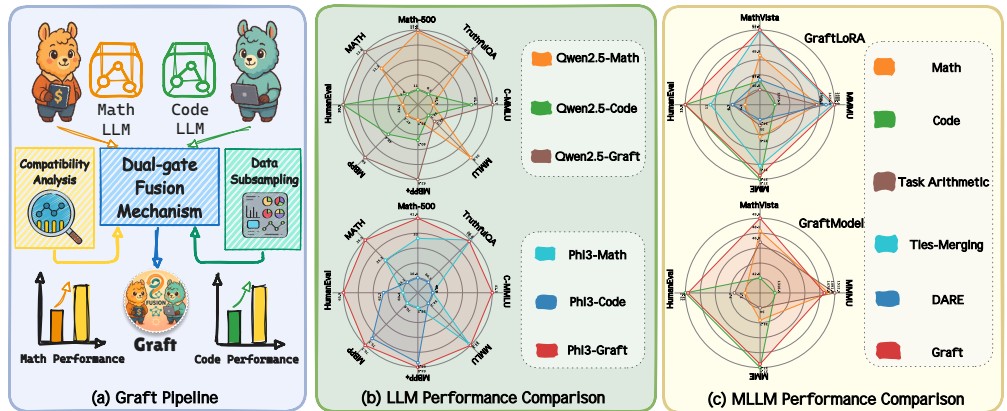

Figure 1: (a) shows the pipeline of our method Graft. (b) and (c) illustrates Graft performance on LLMs and MLLMs compared to other baseline methods.

To address these challenges, we propose a novel parameter fusion method named **Graft**, aiming for more precise and efficient integration of parameters from multiple fine-tuned models. The **GraftModel** variant handles fusion of fully fine-tuned model parameters, while the **GraftLoRA** variant handles fusion of LoRA-adapted model parameters. This dual capability enables flexible knowledge integration from both standard fine-tuned models and LoRA-adapted models. At the local scale, Graft employs a learnable parameter network to measure channel-wise differences, assigning fine-grained weights based on parameter significance. At the global scale, we introduce an entropy-based evaluation mechanism that adjusts fusion weights according to overall parameter information entropy. By synergistically combining local and global assessments through a nonlinear adaptive strategy, Graft effectively mitigates the inherent limitations of conventional linear fusion methods.

Moreover, we further ensure fusion performance by applying a data subsampling methods to prepare representative subsets and evaluating model compatibility through an activation-based compatibility analysis on mismatched datasets. Such methods provide crucial insights, significantly improving fusion decision reliability.

Our contributions are summarized as follows: (1) We present a novel local-global fusion framework that can either merge fully fine-tuned models or LoRA-tuned adapters, enabling precise evaluation and effective integration of diverse model parameters; (2) We introduce a learnable parameter network to capture intricate local differences, substantially enhancing fusion accuracy, and a dynamic entropy-based weighting mechanism enhancing adaptability and generalization; (3) We present a novel single-dataset activation-based compatibility analysis to bolster the reliability of model fusion decisions. (4) We develop a semantic-aware data subsampling technique using K-Means clustering on embeddings to construct diverse and representative datasets, ensuring the robustness of fusing the specialist models.

Collectively, these innovations position Graft as a highly efficient and adaptive parameter fusion method, contributing meaningful theoretical advancements and practical tools that substantially elevate the generalization performance and real-world applicability of large language models.

## 2 RELATED WORK

**Foundation Model Fine-tuning.** The development of AI has transferred deep learning with small models (Zhong et al., 2016; Zhang et al., 2022; Lai, 2019; Zhang et al., 2023) to large language models (LLMs). LLMs acquire domain-specific expertise through Supervised Fine-Tuning (SFT), which adapts pre-trained models to excel in targeted domains. To maintain their original, general capabilities while instilling specialized knowledge, a hybrid strategy interleaves a controlled fraction of general-domain data into the fine-tuning corpus (Que et al., 2024). SFT methodologies can be divided into two paradigms based on parameter-update mechanisms: Full Fine-Tuning, which updates all model parameters and is most effective when abundant data and computational resources are available (Devlin et al., 2019; Radford et al., 2018), and Parameter-Efficient Fine-Tuning (PEFT),

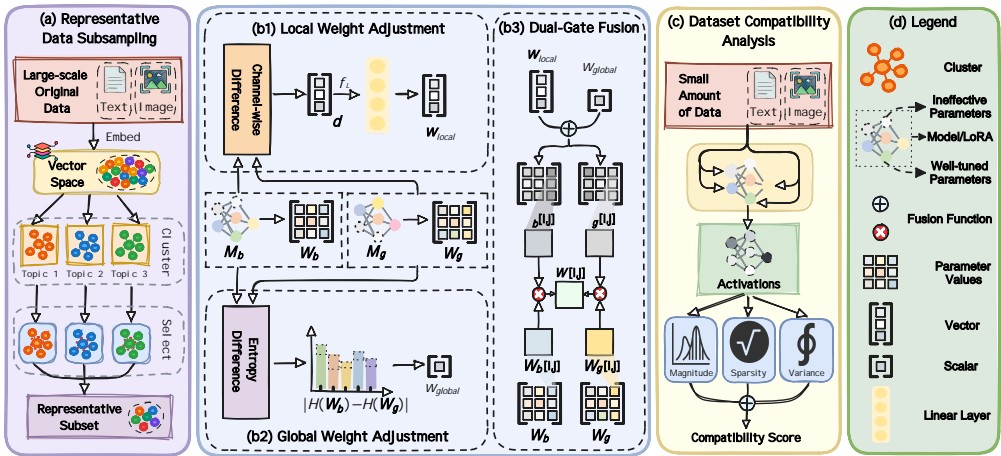

Figure 2: (a) describes the of representative data subsampling methodology. (b) shows the overview of the proposed Graft method, illustrating how base and graft model or LoRA module parameters are fused via a dual-gate fusion mechanism. (c) illustrates the analysis of dataset compatibility.

which freezes the majority of pre-trained weights and updates only a minimal set of additional parameters, thereby drastically reducing both computational cost and storage requirements (Hu et al., 2022; Lester et al., 2021; Liu et al., 2021).

**Model Merging.** Domain model merging techniques build cross-domain models by integrating parameters, bypassing computationally intensive GPU retraining. Early studies (Gupta et al., 2020; Wortsman et al., 2022) employed parameter averaging strategies that simply computed the arithmetic mean of model weights. Although this approach demonstrated moderate performance improvements in multi-domain tasks, it failed to address the varying significance of parameters across domains. Subsequent research introduced mechanisms to evaluate parameter importance, such as Fisher Merging (Matena & Raffel, 2022) and RegMean (Jin et al., 2022). Nevertheless, these techniques exhibit high computational complexity, limiting their widespread adoption. More recently, lightweight fusion paradigms have attracted considerable attention. Task Arithmetic (Ilharco et al., 2022) decomposes the fine-tuning process into additive "task vectors" represented by the difference between the pre-trained model parameters and fine-tuned parameters. Similarly, Ties-Merging (Yadav et al., 2023) alleviates inter-task conflicts through parameter pruning and sign alignment techniques; however, its reliance on global merging coefficients limits fine-grained task-specific adaptability.

**Large Language Models.** The advent of Large Language Models (LLMs), powered by the Transformer architecture, has revolutionized natural language processing. Foundational models like GPT-3 (Floridi & Chiriatti, 2020), LLaMA (Touvron et al., 2023), and T5 (Raffel et al., 2020) have demonstrated remarkable capabilities across a vast spectrum of tasks, including text generation, code completion, and complex reasoning. These models are often pre-trained on massive, general-domain corpora to acquire a broad base of world knowledge. However, the high computational costs of training remain a significant obstacle to the widespread deployment of these large-scale models (Kaplan et al., 2020; Hoffmann et al., 2022). While general-purpose LLMs possess broad knowledge, achieving state-of-the-art performance in specific domains like mathematics (Yang et al., 2024a) or medicine (Tu et al., 2024) requires extensive resources for fine-tuning. Model merging techniques address this challenge by integrating multiple specialized domain models to efficiently build more versatile and capable models, significantly reducing the computational resources required compared to training models from scratch.

## 3 METHODOLOGY

Our approach aims to integrate two distinct modules—*base* and *graft*—into a unified, parameter-efficient module. To achieve this integration, we propose a dual-gate fusion mechanism that simultaneously leverages **local, channel-level** discrepancies (Section 3.1) and **global, distribution-level** divergences of the parameters (Section 3.2), thereby enabling adaptive and informed parameter selection. Moreover, to improve the reliability of fusion decisions across different datasets, we further

introduce dataset compatibility analysis and representative data subsampling, which measures a model's suitability for fusion using an activation-based compatibility metric (Section 3.4) and ensures that our domain-specific models are trained on diverse and high-quality datasets (Section 3.5).

## 3.1 LOCAL WEIGHT ADJUSTMENT

To leverage the strengths of each module on a per-feature basis, we first propose a fine-grained **local weight adjustment** mechanism that dynamically decides, *for each output channel*, whether to emphasize the base module or the graft module. The local weight adjustment mechanism focuses on *channel-wise differences* between these modules. Intuitively, if the two modules differ significantly in a particular output channel, it indicates that they contribute different information for that channel. Therefore, we quantify this difference using the absolute difference between the modules' parameters and use it to guide channel-specific gating decisions.

Let $\mathbf{W}_b \in \mathbb{R}^{M \times N}$ and $\mathbf{W}_g \in \mathbb{R}^{M \times N}$ represent the weight matrices (or flattened parameter sets) of the base and graft modules (e.g., low-rank adaptation layers) for a given layer, where $M$ is the number of output channels (neurons) and $N$ is the number of input features. We measure the absolute difference between $\mathbf{W}_b$ and $\mathbf{W}_g$ for each output channel $i$ as follows:

$$d_i = \sum_{j=1}^{N} |\mathbf{W}_b[i,j] - \mathbf{W}_g[i,j]|, \mathbf{d} = (d_1, d_2, ..., d_M)^\top \in \mathbb{R}^M, \tag{1}$$

This yields a difference vector $\mathbf{d} = (d_1, d_2, \ldots, d_M)^\top \in \mathbb{R}^M$, where each element $d_i$ captures the total absolute deviation between the two modules' weights in channel $i$. A larger $d_i$ implies that the base and graft adapters disagree more in the $i$th channel (i.e., one adapter has learned significantly different feature importance for that channel than the other). Next, we feed this difference vector into a learnable **channel-level gating network**, denoted as $\phi(\cdot)$. The gating network $\phi$ is designed to transform the raw differences $\mathbf{d}$ into an informative gating signal. In practice, $\phi$ could be a small fully-connected module or an affine transformation that processes $\mathbf{d}$ (or each $d_i$ independently) and outputs a corresponding set of gating logits. We then apply a sigmoid activation $\sigma(\cdot)$ to obtain a normalized weight between 0 and 1 for each channel:

$$\mathbf{w}_{local} = \sigma(\phi(\mathbf{d})) \in (0,1)^M, \tag{2}$$

where $\sigma$ represents the sigmoid activation. This finally produces differentiable channel-wise gating weights $\mathbf{w}_{local}$ that emphasize essential parameters.

## 3.2 GLOBAL WEIGHT ADJUSTMENT

We further introduce a **global weight adjustment** mechanism based on the *overall distribution* of the modules' parameters. By comparing distributional characteristics of $\mathbf{W}_b$ and $\mathbf{W}_g$, this mechanism provides a single scalar gating value, determining which module is generally more informative or confident, guiding the fusion at a macro level.

Our approach uses the concept of *entropy* to quantify the distributional characteristics of each module's parameters, which is rooted in fundamental Information Theory. In Shannon's original framework, the entropy of a probability distribution directly measures its information content (Shannon, 1948). In our approach, greater entropy implies that the expert model utilizes a broader spectrum of its parameter space to encode information, rather than collapsing into a sparse or deterministic state which may indicate overfitting to limited features. Furthermore, aligning with Jaynes' Principle of Maximum Entropy, a distribution with greater entropy represents the 'least biased' estimate under given constraints (Jaynes, 1957). By prioritizing experts with higher weight entropy, we focus on modules that maintain the capacity for richer information representation, avoiding unwarranted biases. Specifically, we discretize the parameters into $n$ uniform bins to compute the entropy:

$$H(\mathbf{W}) = -\sum_{k=1}^{n} p_k \log p_k, \; p_k = \frac{|w \in \mathbf{W}|w \in B_k|}{M \times N} \tag{3}$$

where the numerator is the number of elements of $\mathbf{W}$ whose value lies in the interval defining bin $B_k$, and $M \times N$ is the total number of parameters in $\mathbf{W}$. Based on the entropy difference between

base and graft adapters, we determine a global fusion scalar weight:

$$w_{global} = \frac{a}{c}\arctan(c[H(\mathbf{W}_b) - H(\mathbf{W}_g)]) + \frac{1}{2} \in (0, 1), \tag{4}$$

where $a$ and $c$ are constants that shape the $\arctan$ function's output range and slope. Here, $w_{global}$ is a scalar constrained to $(0, 1)$, serving as a global gating factor. In summary, the global weight adjustment encapsulates a high-level judgment of which module appears to carry more information content in its parameters.

## 3.3 DUAL-GATE FUSION STRATEGY

The final fusion incorporates both local and global gating weights to construct comprehensive fusion weights:

$$\tilde{\mathbf{w}}_b = w_{global} \cdot \frac{\exp(w_{global} \cdot \mathbf{w}_{local}) - 1}{\exp(w_{global} \cdot \mathbf{w}_{local})}, \tag{5}$$

$$\tilde{\mathbf{w}}_g = (1 - w_{global}) \cdot \frac{\exp((1 - w_{global}) \cdot (1 - \mathbf{w}_{local})) - 1}{\exp((1 - w_{global}) \cdot (1 - \mathbf{w}_{local}))}. \tag{6}$$

These intermediate weights are normalized using softmax to ensure stable and adaptive fusion across all parameter channels:

$$[w_b, w_g] = \text{Softmax}([\tilde{w}_b, \tilde{w}_g]), \tag{7}$$

$$\mathbf{W}_{fused} = w_b \odot \mathbf{W}_b + w_g \odot \mathbf{W}_g. \tag{8}$$

This fusion strategy explicitly captures and resolves parameter-level conflicts while optimizing overall model generalization and adaptation capabilities. The overall strategy is summarized as Algorithm 1:

---

**Algorithm 1** Graft Fusion

---

**Require:** Base $W_b \in \mathbb{R}^{M \times N}$, graft $W_g \in \mathbb{R}^{M \times N}$, gate net $\phi$, scalars $a, c$
**Ensure:** Fused $W_f$
1: $d \leftarrow \sum_j |W_b - W_g|$
2: $D \leftarrow \text{expand}(d)$
3: $w_{\text{loc}} \leftarrow \sigma(\phi(D))$
4: $H_b \leftarrow \text{entropy}(W_b)$, $H_g \leftarrow \text{entropy}(W_g)$
5: $w_{\text{glob}} \leftarrow \frac{a}{c}\arctan(c(H_b - H_g)) + \frac{1}{2}$
6: $\tilde{w}_b \leftarrow w_{\text{glob}}(\frac{\exp(w_{glob} \cdot \mathbf{w}_{loc}) - 1}{\exp(w_{glob} \cdot \mathbf{w}_{loc})})$
7: $\tilde{w}_g \leftarrow (1 - w_{\text{glob}})(\frac{\exp((1 - w_{glob}) \cdot (1 - \mathbf{w}_{loc})) - 1}{\exp((1 - w_{glob}) \cdot (1 - \mathbf{w}_{loc}))})$
8: $[w_b, w_g] \leftarrow \text{Softmax}([\tilde{w}_b, \tilde{w}_g])$
9: $W_f \leftarrow w_b \odot W_b + w_g \odot W_g$

---

## 3.4 DATASET COMPATIBILITY ANALYSIS

In the practice of fusion, selecting appropriate domain-specific models is a crucial step. Models suitable for the target dataset domain can provide a strong starting point for fusion; conversely, mismatched models can even lead to the degradation of the fused model. For more reasonable selection of models, we propose an analysis method to assess dataset compatibility for fully fine-tuned models or LoRA-adapters fusion at the module level. This analysis introduces an activation-based metric - *compatibility*, indicating the suitability for the given dataset.

Specifically, we choose $K$ input samples from the target dataset, where $K$ is a relatively small value comparing to the total number of samples in the target dataset. Let the activations be $\mathbf{A}_i^{(k)} \in \mathbb{R}^{B \times D}$, where $B$ is the batch size, $D$ the activation dimension, $i$ indexes modules, and $k$ indexes samples.

From these activations we compute three statistics per module:

$$\text{Mean magnitude}: \quad \mu_i = \frac{1}{K} \sum_{k=1}^{K} \frac{\|\mathbf{A}_i^{(k)}\|_1}{\dim(\mathbf{A}_i^{(k)})}, \tag{9}$$

$$\text{Sparsity}: \quad s_i = \frac{1}{K} \sum_{k=1}^{K} \frac{\#\{j : |(\mathbf{A}_i^{(k)})_j| < \epsilon\}}{\dim(\mathbf{A}_i^{(k)})}, \tag{10}$$

$$\text{Variance}: \quad v_i = \frac{1}{K} \sum_{k=1}^{K} \text{Var}(\mathbf{A}_i^{(k)}). \tag{11}$$

Based on these metrics, a comprehensive *data sensitivity* score is computed:

$$\rho_i = \mu_i \times (1 - s_i) \times \sqrt{v_i} \tag{12}$$

which quantifies the module's sensitivity to the given dataset. Higher sensitivity scores reflect stronger engagement of the module's parameters, indicating favorable compatibility for fusion. Moreover, we perform global min-max normalization across modules for each metric, yielding normalized scores $\mu_i'$, $s_i'$, and $v_i'$, enhancing comparability across modules. The normalized sensitivity is then calculated as:

$$\rho_i' = \mu_i' \times (1 - s_i') \times \sqrt{v_i'} \tag{13}$$

Finally, compatibility across all modules is summarized into an aggregate metric:

$$\text{compatibility} = \frac{1}{M} \sum_{i=1}^{M} \rho_i' \tag{14}$$

where $M$ represents the total number of evaluated modules. This metric serves as a criterion for evaluating model suitability. In practice, Modules with compatibility exceeding the threshold are considered acceptable for fusion. The utilization of this compatibility metric effectively improves the quality of model fusion and subsequent downstream performance.

### 3.5 REPRESENTATIVE DATA SUBSAMPLING

To ensure that the domain-specific models are fused on high-quality, diverse data while adhering to a controlled computational budget, we introduce a structured data subsampling methodology. This approach moves beyond simple random sampling, which can lead to skewed distributions where common topics are over-represented. Instead, our method focuses on preserving the semantic diversity of the original large-scale corpora. The process consists of three main stages:

**1. Semantic Embedding.** We first transform each data instance (e.g., a question-answer pair or an instruction-response pair) into a high-dimensional vector representation using a pre-trained sentence transformer model. This step maps the semantic content of the data into a continuous vector space, where proximity between vectors corresponds to similarity in meaning.

**2. K-Means Clustering.** With the data represented as embeddings, we apply the K-Means algorithm to partition the entire embedding space. To determine the optimal number of clusters $K$, we avoid arbitrary selection by employing an automated Silhouette Analysis. We iterate through a range of potential $K$ values, calculating the Silhouette Score for each to measure the trade-off between intra-cluster cohesion and inter-cluster separation. We select the $K$ that maximizes this score, ensuring that the resulting clusters accurately reflect the intrinsic semantic structure of the specific dataset.

**3. Centroid-Based Selection.** Finally, we construct the representative subset by sampling instances from each cluster. For each cluster, we select the data points that are closest to its geometric centroid. This centroid-based approach ensures that the chosen samples are the most prototypical examples of each semantic group, thereby preserving the breadth and diversity of the original dataset in a compact and balanced form.

Table 1: Comparison results of LLM performances on domain-specific tasks (Math-500, MATH, HumanEval, MBPP, MBPP+) and general benchmarks (MMLU, TruthfulQA) after applying Graft. We use **bold** text to indicate the best results for each model on each benchmark.

| Model | Math-500 | MATH | HumanEval | MBPP | MBPP+ | MMLU | C-MMLU | TruthfulQA |
|---|---|---|---|---|---|---|---|---|
| Qwen2-1.5B-Math (Yang et al., 2024a) | 21.2 | **18.8** | 34.8 | 45.8 | 38.9 | 50.8 | **61.4** | 6.4 |
| Qwen2-1.5B-Code | 12.6 | 7.9 | 39.6 | **51.3** | 42.3 | 49.9 | 60.2 | 2.8 |
| Qwen2-1.5B-Math-Code | **22.8** | 18.2 | **40.2** | 50.8 | **44.2** | **50.9** | 61.2 | **8.0** |
| Qwen2.5-0.5B-Math (An et al., 2024) | 17.6 | 11.3 | 28.1 | 47.1 | 39.7 | **34.8** | 42.5 | 0.9 |
| Qwen2.5-0.5B-Code | 11.0 | 7.4 | **30.5** | 48.4 | 40.7 | 34.1 | 45.2 | 0.1 |
| Qwen2.5-0.5B-Math-Code | **17.8** | **13.8** | **30.5** | 50.0 | 42.6 | 34.2 | **46.7** | **1.0** |
| Phi-3-4B-Math (Abdin et al., 2024) | 33.0 | 25.3 | 60.4 | 70.6 | 58.2 | 57.4 | 40.9 | **76.4** |
| Phi-3-4B-Code | 16.4 | 14.1 | 62.2 | 75.7 | 64.3 | 54.2 | 40.9 | 66.1 |
| Phi-3-4B-Math-Code | **41.4** | **35.5** | **65.9** | **76.7** | **64.8** | **57.5** | **41.3** | 76.1 |

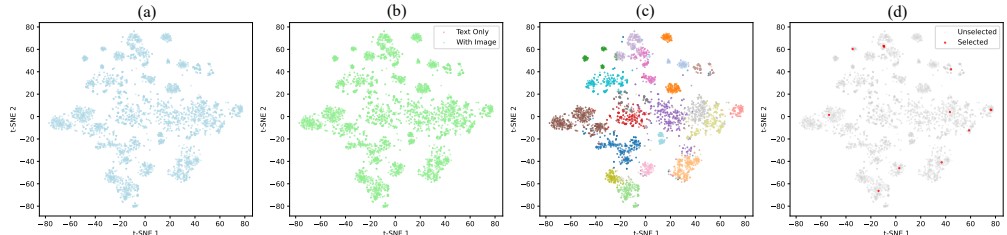

Figure 3: t-SNE visualization of the experimental data preparation. This figure illustrates (a) the original semantic landscape of the dataset, (b) the data distribution by modality type, (c) the $K$-Means clustering to categorize diverse topics, and (d) the selected representative samples (red dots).

## 4 EXPERIMENTS

### 4.1 DATA AND EXPERIMENTAL SETUP

**Data Details.** To evaluate the model's cross-domain generalization under a controlled data budget, we select 1,000 to 5,000 instances from publicly available corpora using the representative data subsampling method detailed in Section 3.5. The corpora are: MetaMathQA-40K (Yu et al., 2023a), Code-Instructions-120K-Alpaca (iamtarun, 2023), MathV-360K (Shi et al., 2024), PathVQA (He et al., 2020), Sujet-Finance-QA-Vision-100K (AI, 2024), and Code-Alpaca-20K (Chaudhary, 2023). The visual outcome of this data subsampling process is illustrated in Figure 3.

**Experimental Setup.** We conduct experiments on the Qwen2-1.5B (Yang et al., 2024a), Qwen2.5-0.5B (An et al., 2024), Phi-3-4B (Abdin et al., 2024), and Qwen2-VL-2B (Wang et al., 2024a) models. For the hyperparameters in Graft modules, we set the global gating adjustment parameters $a = 0.4$ and $c = 500$. The entropy calculation uses $n = 10$ bins for discretizing weight distributions. All experiments were conducted on A6000 GPUs, using the same hyperparameter settings across all domain adaptation scenarios to ensure fair comparison.

### 4.2 EXPERIMENTAL RESULTS

**Overall Performance Comparison.** We fused domain-specific models fine-tuned for mathematics and coding and assessed their performance on a range of specialized and general benchmarks, with the results presented in Table 1. The findings show that Graft effectively integrates knowledge from specialized experts. For instance, the fused Qwen2-1.5B model achieves the highest scores on both the math benchmark Math-500 (22.8) and code benchmarks like HumanEval (40.2) and MBPP+ (44.2), demonstrating a successful combination of capabilities. This trend is even more pronounced with the Phi-3-4B. It scores 41.4 on Math-500, surpassing the math expert (33.0), and 65.9 on HumanEval, exceeding the code expert (62.2). Furthermore, across all model families, the fused versions maintain or slightly improve performance on general benchmarks like MMLU and TruthfulQA. This confirms that our fusion process successfully integrates specialized skills without causing catastrophic forgetting of the models' general knowledge base.

Table 2: Comparison results of MLLM performances (Full vs LoRA) on domain-specific tasks (MathVista, HumanEval) and general benchmarks (MMMU, MME). We use **bold** text to indicate the best results and underline to indicate the second-best results.

| Model | GraftModel Performance | | | | GraftLoRA Performance | | | |
|---|---|---|---|---|---|---|---|---|
| | MathVista$^F$ | HumanEval$^F$ | MMMU$^F$ | MME$^F$ | MathVista$^L$ | HumanEval$^L$ | MMMU$^L$ | MME$^L$ |
| Qwen2-VL-2B (Wang et al., 2024b) | 47.8 | 14.0 | 34.6 | 1473.5 | 47.8 | 14.0 | 34.6 | 1473.5 |
| Math | 48.1 | 1.2 | 34.7 | 1491.9 | 49.9 | 4.3 | 35.6 | 1455.7 |
| Code | 42.6 | **15.2** | **37.4** | 1316.5 | 47.6 | **15.9** | **37.8** | 1373.9 |
| Task Arithmetic (Ilharco et al., 2022) | 46.8 | 3.7 | 34.0 | **1512.9** | 46.8 | 6.7 | 35.0 | 1454.0 |
| Ties-Merging (Yadav et al., 2023) | 48.3 | 6.7 | 35.0 | 1472.3 | 52.1 | 11.0 | 37.1 | 1484.1 |
| DARE (Yu et al., 2024) | 49.5 | 10.4 | 36.2 | 1492.6 | 47.7 | 6.7 | 34.7 | 1471.9 |
| **Our Method** | **49.6** | 14.6 | 37.2 | 1478.9 | **52.2** | **15.9** | 37.6 | **1488.4** |

Table 3: Performance of single-domain and fused models across multiple domains. (✓ indicates the domain(s) included in the model).

| Domain Composition | | | | Compatibility Scores | | Benchmark Scores | | | |
|---|---|---|---|---|---|---|---|---|---|
| Math | Code | Fin. | Med. | Math | Code | MathVista | HumanEval | MMMU | MME |
| ✓ | | | | 0.331 | – | 49.9 | 4.3 | 35.6 | 1455.7 |
| | ✓ | | | – | 0.286 | 47.6 | 15.9 | 37.8 | 1373.9 |
| | | ✓ | | – | – | 43.8 | 8.5 | 36.7 | 1414.7 |
| | | | ✓ | – | – | 46.0 | 12.2 | 37.5 | 1470.2 |
| ✓ | ✓ | | | 0.282 | 0.204 | 52.2 | 15.9 | 37.6 | 1488.4 |
| ✓ | | ✓ | | 0.280 | – | 50.1 | – | 37.3 | 1470.1 |
| ✓ | | | ✓ | 0.315 | – | **52.4** | – | 37.0 | **1535.0** |
| | ✓ | ✓ | | – | 0.182 | – | **16.5** | **38.1** | 1457.5 |
| | ✓ | | ✓ | – | 0.155 | – | **16.5** | 38.0 | 1468.6 |

**Adaptability of Multimodal Models.** Fine-tuning MLLMs is often more computationally intensive and time-consuming than for LLMs due to complex vision-language data and larger model architectures. This makes a data-efficient merging technique like Graft particularly valuable, since the core principles of our dual-gate fusion mechanism are not inherently limited to multimodality. To validate this, we conduct experiments on multimodal language models. Table 2 summarizes the cross-domain performance of our fusion strategy on four multimodal benchmarks: MathVista, HumanEval, MMMU, and MME. Compared with the pretrained backbone Qwen2-VL-2B and three competitive weight-merging baselines (Task Arithmetic(Ilharco et al., 2022), Ties-Merging(Yadav et al., 2023) and DARE(Yu et al., 2024)), the proposed Graft delivers the most balanced improvements.[1] The superscripts F and L in Table 2 denote fully fine-tuned and LoRA fine-tuned models, respectively. Notably, across all fusion scenarios, the LoRA-tuned domain experts consistently outperform their fully fine-tuned counterparts. For example, fusing LoRA-based adapters yields a MathVista accuracy of 52.2% compared to 49.6% with full fine-tuning, and similarly improves the HumanEval pass@1 from 14.6% to 15.9%. This trend holds across all evaluated methods, indicating that LoRA preserves complementary knowledge more effectively for model merging.

**Cross-Domain Compatibility Analysis.** Table 3 extends the compatibility-sensitive fusion analysis beyond the Math–Code pair reported in Table 2 by evaluating additional cross-domain settings. Notably, Math+Medical achieves the highest MathVista score (52.4), while Code+Finance yields the best HumanEval accuracy (16.5), indicating that Graft is applicable for complementary knowledge across heterogeneous domains. The analysis of the compatibility scores based on activation in Table 3 further substantiates their predictive value for the fusion of domains. For Math-centric pairs, higher scores like Math+Medical (0.314) correlate with larger performance improvements. An apparent outlier arises in the Code + Medical case: despite a modest score (0.155), the fused model still excels on HumanEval. This behaviour is attributable to the Medical expert's already competitive baseline on that task, which narrows the observable gain. Consequently, the compatibility score is most informative when considered alongside baseline performance. We therefore recommend a two-factor rule for selecting fusion pairs, weighing both: (i) the activation compatibility score and (ii) the stronger expert's standalone performance on the target benchmark.

---

[1] All methods considered for comparison in this study are fully open-source; closed-source or commercial systems are excluded to ensure reproducibility.

Table 4: Results of multi-domain fusion (✓ indicates included domain).

| Domain Composition | | | | Benchmark Scores | |
|---|---|---|---|---|---|
| Math | Code | Finance | Medical | MathVista | HumanEval |
| ✓ | ✓ | ✓ | | 51.7 | 14.6 |
| ✓ | ✓ | | ✓ | 52.9 | 14.6 |
| ✓ | ✓ | ✓ | ✓ | 53.0 | 14.6 |

Table 5: Ablation study on gating components (✓ indicates enabled part).

| Local | Global | Benchmark Scores | | | |
|---|---|---|---|---|---|
| | | MathVista | HumanEval | MMMU | MME |
| ✓ | | 52.0 | 15.9 | 37.6 | 1495.0 |
| | ✓ | 51.7 | 12.2 | 37.6 | 1483.4 |
| ✓ | ✓ | 52.2 | 15.9 | 37.6 | 1488.4 |

Table 6: Ablation study on Entropy vs. Spectral Norm (✓ indicates enabled part).

| Model | Spectral | Entropy | Math-500 | MATH | HumanEval | MBPP | MBPP+ | MMLU | C-MMLU | TruthfulQA |
|---|---|---|---|---|---|---|---|---|---|---|
| Qwen2-1.5B-Math-Code | ✓ | | 20.6 | **18.7** | **40.8** | 50.0 | 42.6 | 50.5 | **61.3** | 6.6 |
| Qwen2-1.5B-Math-Code | | ✓ | **22.8** | 18.2 | 40.2 | **50.8** | **44.2** | **50.9** | 61.2 | **8.0** |
| Qwen2.5-0.5B-Math-Code | ✓ | | **18.0** | 13.1 | 28.1 | 49.5 | 42.1 | 34.1 | **46.7** | 0.2 |
| Qwen2.5-0.5B-Math-Code | | ✓ | 17.8 | **13.8** | **30.5** | **50.0** | **42.6** | **34.2** | **46.7** | **1.0** |
| Phi-3-4B-Math-Code | ✓ | | 39.2 | 34.0 | 64.6 | 74.1 | 61.9 | 54.6 | **41.3** | 74.5 |
| Phi-3-4B-Math-Code | | ✓ | **41.4** | **35.5** | **65.9** | **76.7** | **64.8** | **57.5** | **41.3** | **76.1** |

**Multi-Domain Fusion.** We next evaluate the scalability of Graft to multi-domain integration by fusing three and four expert adapters, with results in Table 4. Adding each new expert yields diminishing yet still positive gains on MathVista: fusing Math + Code with the Finance adapter results in an accuracy of 51.7, while substituting Medical further boosts it to 52.9. Integrating all four domains reaches 53.0, delivering a 0.6-point absolute improvement over the best two-domain model. These monotonic gains indicate that heterogeneous domain knowledge compounds to benefit mathematical reasoning. Coding performance, measured by HumanEval, remains constant (14.6) as additional domains are grafted. Although this is slightly below the two-domain peak (15.9), the negligible drop confirms that our dual-gating mechanism effectively suppresses interference from unrelated experts. These findings demonstrate that our framework scales gracefully beyond pairwise fusion, unifying multiple adapters without catastrophic forgetting and showing its promise for constructing broadly capable LLMs.

**Ablation Study on Gating Components.** Table 5 compares three gating schemes: Local-Gate, Global-Gate, and Dual-Gate. Dual-Gate consistently outperforms its single-gate counterparts, achieving 52.2 on MathVista (vs. Local-Gate: 52.0 and Global-Gate: 51.7), 15.9 on HumanEval (vs. Local-Gate: 15.9 and Global-Gate: 12.2), 37.6 on MMMU (on par), and a 1488.4 composite score on MME. Mechanistically, Local-Gate amplifies fine-grained, domain-specific signals with a channel-wise mask. In contrast, Global-Gate balances cross-domain knowledge coarsely using a single weight derived from entropy. Dual-Gate synergistically combines these perspectives: the local gate preserves micro-features while the global gate mitigates inter-domain conflicts. This complementary interaction enables the model to retain specialized expertise without sacrificing holistic performance, which explains the superior results observed on all metrics in Table 5.

**Ablation Study on Entropy vs. Spectral Norm.** To rigorously validate our choice of the entropy-based gating mechanism, we perform an additional ablation study comparing it against a Spectral Norm-based strategy. In the spectral variant, we utilize the spectral norm of the weight matrices to calculate the global gating scalar, replacing the entropy metric. As presented in Table 6, the Entropy-based approach demonstrates superior performance in the majority of cases. For instance, on the Phi-3-4B model, the Entropy strategy consistently outperforms the Spectral method across all five benchmarks, achieving a notable margin on Math-500 (41.4 vs. 39.2). While the Spectral method shows marginal gains in specific isolated cases (e.g., Qwen2-1.5B on MATH), the Entropy-based mechanism offers a more robust and generalized estimation of parameter importance for cross-domain fusion.

**Scalability to Larger Models.** To demonstrate the scalability of Graft beyond smaller parameter scales, we extend our experiments to larger foundation models, specifically Qwen2.5-7B and Llama-3-8B. We apply our fusion strategy to integrate Math and Code experts derived from these architectures. Table 7 illustrates the performance of the individual models compared to the fused Graft model. For Qwen2.5-7B, the fused model (Math+Code) achieves a score of 64.2 on Math-500, improving upon the specialized Math expert (62.6), while simultaneously preserving the coding capability (77.4 on HumanEval) relative to the Code expert. Similarly, for Llama-3-8B, the fused model demonstrates

Table 7: Scalability to larger models.

| Model | Math-500 | MATH | HumanEval |
|---|---|---|---|
| Qwen2.5-7B-Math (Qwen et al., 2025) | 62.6 | 51.7 | 70.7 |
| Qwen2.5-7B-Code | 21.6 | 23.5 | **79.3** |
| Qwen2.5-7B-Math-Code | **64.2** | **54.2** | 77.4 |
| Llama-3-8B-Math (Grattafiori et al., 2024) | 25.6 | **18.8** | 62.4 |
| Llama-3-8B-Code | 21.6 | 14.5 | 65.6 |
| Llama-3-8B-Math-Code | **26.0** | 17.8 | **67.2** |

Table 8: Merging efficiency analysis.

| Model | Time Cost (s) | | |
|---|---|---|---|
| | Math | Code | Math-Code |
| Qwen2-1.5B | 15089.7 | 29591.3 | 4576.2 |
| Qwen2.5-0.5B | 7852.9 | 18608.2 | 3932.6 |
| Qwen2.5-7B | 51051.2 | 79770.0 | 5097.8 |
| Phi3-4B | 29755.1 | 68195.5 | 3612.1 |
| Llama-3-8B | 39958.5 | 76791.1 | 5742.3 |

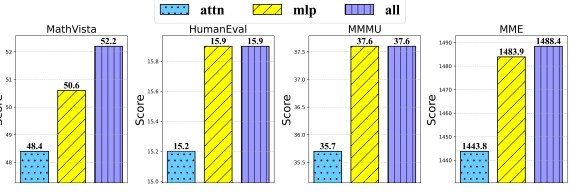

Figure 4: Performance of different projection layer fusion strategies.

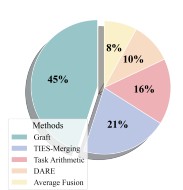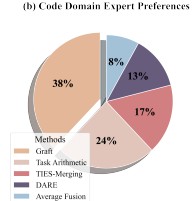

Figure 5: Human preference for generated content of baselines and our model.

positive transfer even at larger scales, outperforming the individual experts on both HumanEval (67.2) and Math-500 (26.0). These results confirm that Graft's dual-gate mechanism effectively scales to larger architectures without suffering from catastrophic interference.

**Merging Efficiency Analysis.** We evaluate the computational cost (in seconds) of training distinct domain experts versus the cost of the Grafting process. As shown in Table 8, the Graft process requires only a small fraction of the time needed to train a single domain expert. For example, grafting the Qwen2.5-7B experts takes approximately 5,742 seconds, while training the individual Code expert requires over 76,000 seconds. This confirms the effectiveness of our approach in rapidly creating multi-domain models from existing experts with insignificant computational overhead.

**Layer-wise Fusion Analysis.** To investigate parameter fusion granularity, we conduct studies on selectively merging different projection layers in Transformer blocks. As shown in Figure 4, we compare three fusion strategies: (1) attn: merging only attention projections; (2) mlp: merging only MLP projections; (3) all: jointly merging both. Merging all projection layers attains 52.2 on MathVista (+3.8 over "attn", +1.6 over "mlp") and 15.9 on HumanEval (+0.7 over "attn"), indicating synergistic benefits from cross-module knowledge integration. The results demonstrate that comprehensive layer fusion achieves optimal performance across all benchmarks. This validates our design choice of full-layer fusion, which maximizes the preservation of both structural relationships (via attention) and feature representations (via MLPs).

**Human Evaluation.** We further conduct an expert preference study to evaluate the effectiveness of Graft across domains. We recruited ten domain experts (5 mathematics, 5 computer science) to rank the responses from the fused models of five fusion methods (Average, Task Arithmetic, TIES-Merging, DARE, and Graft) on randomly sampled queries from MathVista and HumanEval. As shown in Figure 5, results demonstrate a clear preference for Graft across both domains. The expert preference results align with our quantitative performances, demonstrating that our dual-gate fusion approach successfully preserves domain-specific knowledge while enabling cross-domain integration.

## 5 CONCLUSION

In this work, we introduce Graft, a dual-gate parameter fusion framework that synergistically combines local channel-level gating with a global entropy-based weighting mechanism to integrate parameters from different domain experts. To ensure the reliability of their fusion, we also propose a representative data subsampling technique for data selection and a single-dataset activation-based compatibility analysis that quantitatively predicts complementary domain pairs prior to merging.

## ETHICS STATEMENT

This work adheres to the ICLR Code of Ethics. Our research did not involve human subjects or animal experimentation. The human evaluation component consisted of domain experts assessing anonymized model outputs for performance analysis, without the collection of any personal data. All datasets used in our experiments, including MetaMathQA, Code-Alpaca-20K, MathVista, and others, are publicly available and were utilized in accordance with their respective licenses and usage guidelines. We acknowledge that the foundational models and public datasets may contain inherent biases, and that the fusion process presents a potential risk of propagating or compounding these biases. Our contributions are purely methodological, focusing on the framework for parameter fusion and its benchmark evaluation. This work does not develop or endorse specific real-world applications of the resulting models, and we encourage thorough safety and fairness assessments for any downstream use.

## REPRODUCIBILITY STATEMENT

The code and data for our project are available at `https://anonymous.4open.science/r/Graft-8213`. Detailed descriptions of hyperparameters and experimental settings can be found in Appendix D.

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

## A    APPENDIX

This is the Appendix for "Graft: Integrating the Domain Knowledge via Efficient Parameter Synergy for LLMs". Table 9 summarizes the abbreviations and symbols used in the main paper. This Appendix

**Table 9:** Abbreviations and symbols used in the main paper.

| Abbreviation/Symbol | Meaning |
| :---: | :---: |
| *Abbreviation* | |
| LLMs | Large Language Models |
| MLLMs | Multimodal Large Language Models |
| CAPS | Compatibility-Aware Parameter Splicing |
| SFT | Supervised Fine-Tuning |
| PEFT | Parameter-Efficient Fine-Tuning |
| SR | Scientific Reasoning |
| TQA | Textbook Question Answering |
| NC | Numeric Commonsense |
| AR | Arithmetic Reasoning |
| VQA | Visual Question Answering |
| GR | Geometry Reasoning |
| ALR | Algebraic Reasoning |
| GPS | Geometry Problem Solving |
| MWP | Math Word Problem |
| LR | Logical Reasoning |
| FQA | Figure Question Answering |
| SRG | Statistical Reasoning |
| MLP | Multi-Layer Perception |
| *Symbol in Algorithm* | |
| $\mathbf{W}_b$ | Base model weight matrix |
| $\mathbf{W}_g$ | Graft model weight matrix |
| $\mathbf{d}$ | Channel-wise absolute difference vector |
| $\phi$ | Learnable channel-level gating network |
| $\sigma$ | Sigmoid activation function |
| $\mathbf{w}_{local}$ | Local gating weights from $\phi$ network |
| $H(\mathbf{W})$ | Entropy function for weight matrices |
| $H_b, H_g$ | Entropy of $\mathbf{W}_b$ and $\mathbf{W}_g$ |
| $a, c$ | Hyperparameters for global weight adjustment |
| $w_{global}$ | Global gating scalar from entropy |
| $\tilde{w}_b, \tilde{w}_g$ | Intermediate fusion weights |
| $w_b, w_g$ | Normalized fusion weights via softmax |
| $\mathbf{W}_{fused}$ | Final fused weight matrix |
| $\mu$ | Mean magnitude |
| $s$ | Sparsity |
| $v$ | Variance |
| $K$ | Numbers of input samples |
| $\mathbf{A}_i^{(k)}$ | Activations |
| $\rho$ | Data sensitivity score |
| $\mu', s', v'\ k', \rho'$ | Normalized scores |

is organized as follows:

- Section B presents the novelty and contribution of our approach, providing a comprehensive comparison with other baseline methods.
- Section C demonstrates extensive additional experimental results across multiple dimensions.
- Section D details the training settings and hyperparameters for both GraftModel and Graft-LoRA implementations.
- Section E provides information on the dataset details used in our experiments.
- Section F offers a comprehensive description of the baseline methods employed in our comparative experiments.

- Section G indicates the use of LLMs in our work.
- Section H discusses the broader impact and future direction of our approach.

## B NOVELTY AND CONTRIBUTION

Our research aims to unlock the full potential of parameter fusion approaches by implementing a dual-gate mechanism that addresses parameter competition across different domains. We re-examine existing model merging methods and highlight the critical role of compatibility-aware parameter integration. To clearly demonstrate the innovation of our method, we conduct a comparative analysis with existing state-of-the-art baseline methods.

**Comparison with Task Arithmetic.** Both Task Arithmetic and our Graft approach aim to integrate knowledge from specialized models. However, there are several key differences:

- Task Arithmetic(Ilharco et al., 2022) employs linear combinations with scalar coefficients, whereas Graft utilizes channel-wise adaptive fusion that captures fine-grained parameter importance.
- When addressing parameter conflicts, Task Arithmetic(Ilharco et al., 2022) lacks a mechanism to evaluate parameter significance, potentially leading to destructive interference when domains have opposing feature preferences.
- Our dual-gate fusion mechanism integrates both local channel-level differences and global entropy-based weighting, enabling more nuanced parameter selection.

**Comparison with TIES-Merging.** Both TIES-Merging(Yadav et al., 2023) and our Graft approach address parameter interference through parameter-level adjustments. However, there are several key distinctions:

- TIES-Merging(Yadav et al., 2023) employs sign-based pruning with binary decisions, neglecting the continuous spectrum of parameter importance. In contrast, Graft considers the significance of fine-grained parameter by quantifying channel differences.
- In terms of cross-domain integration, TIES-Merging(Yadav et al., 2023) only considers parameter sign concordance, whereas our method leverages both functional attribution and information-theoretic signals to guide fusion.
- Our compatibility scoring mechanism quantifies inter-expert alignment at the activation level, providing a principled approach to domain pair selection that TIES-Merging lacks.

**Comparison with DARE.** Both DARE and Graft integrate capabilities from specialized models, but there are significant differences:

- DARE(Yu et al., 2024) applies a mask function to parameter differences without considering domain compatibility, while we employ compatibility analysis to guide parameter fusion.
- DARE(Yu et al., 2024) primarily focuses on preserving the performance of individual experts, whereas Graft specifically targets synergistic integration that enhances cross-domain capabilities.
- Our activation-based compatibility metric provides a quantitative basis for model selection prior to fusion, which DARE(Yu et al., 2024) lacks. This enables more informed fusion decisions, as evidenced by the strong correlation between compatibility scores and performance gains.

## C ADDTIONAL RESULTS

### C.1 EXTENDED EVALUATION ON OTHER MODELS

To further validate the effectiveness and scalability of our approach, we conduct additional experiments on Qwen2.5-VL-3B(Bai et al., 2025) across four specialized domains. Table 10 presents the performance comparison between single-domain models and their fused counterparts across multiple

benchmarks. The Math-specialized model achieves the highest MathVista(Lu et al., 2024) score of 59.4, while the Code-specialized model excels in HumanEval with 29.3. Our fusion strategy demonstrates consistent improvements: the Math&Code combination achieves 59.6 on MathVista (+0.2 improvement) while maintaining competitive HumanEval performance at 27.6.

**Table 10:** Performance of Qwen2.5-VL-3B on single-domain and fused models across multiple domains. (✓ indicates the domain(s) included in the model).

| Domain Composition | | | | Benchmark Scores | |
|---|---|---|---|---|---|
| **Math** | **Code** | **Finance** | **Medical** | **MathVista** | **HumanEval** |
| ✓ | | | | 59.4 | 25.6 |
| | ✓ | | | 58.2 | 29.3 |
| | | ✓ | | 57.2 | 23.2 |
| | | | ✓ | 57.9 | 12.8 |
| ✓ | ✓ | | | 59.6 | 27.6 |
| ✓ | | ✓ | | 57.8 | – |
| ✓ | | | ✓ | 57.9 | – |
| | ✓ | ✓ | | – | 29.3 |
| | ✓ | | ✓ | – | 29.3 |

## C.2 SAMPLE SIZE SELECTION ANALYSIS

To determine the optimal number of samples for our fusion procedure, we conduct a systematic empirical study across varying sample sizes from 1,000 to 5,000. As illustrated in Figure 6 and Figure 7, our systematic evaluation across varying sample sizes reveals distinct convergence patterns for different fusion approaches. For GraftLoRA fusion, the performance trajectory shows that while peak performance on individual benchmarks occurs at intermediate sample sizes (e.g., 3000 samples achieving 52.3% on MathVista), the 5000-sample configuration delivers the most consistent performance across all evaluation metrics, achieving 52.2% on MathVista, 15.9% on HumanEval, 37.6% on MMMU, and 1488.4 on MME.

Similarly, GraftModel demonstrates monotonic performance improvements with increasing sample sizes, reaching optimal performance at 5000 samples with 49.6% on MathVista and 15.2% on HumanEval. This convergence behavior indicates that our selection of sample size provides sufficient statistical representation for effective parameter alignment while avoiding potential overfitting observed in smaller sample regimes.

Figure 6: Performance analysis across training sample sizes on GraftLoRA.

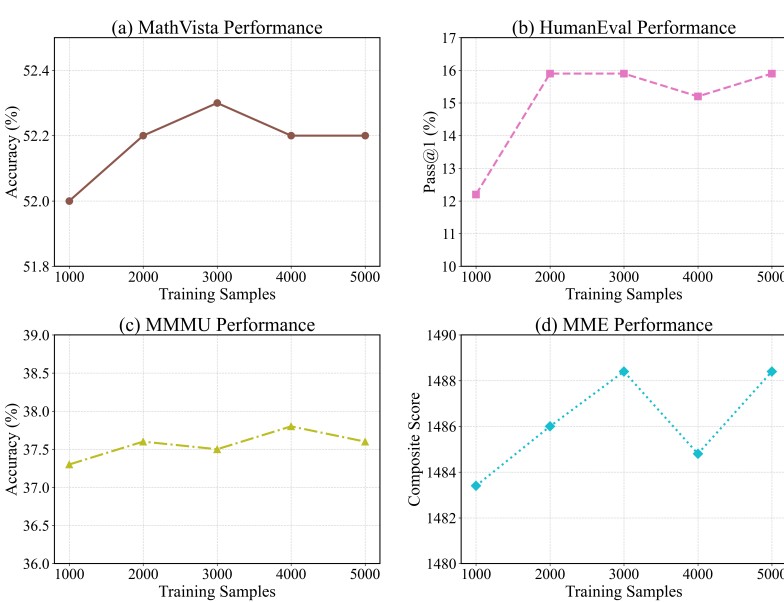

Figure 7: Performance analysis across training sample sizes on GraftModel.

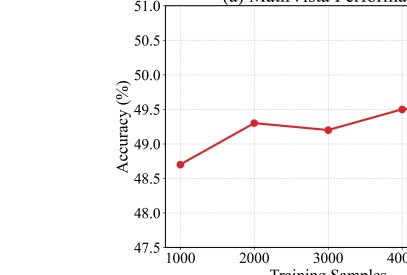
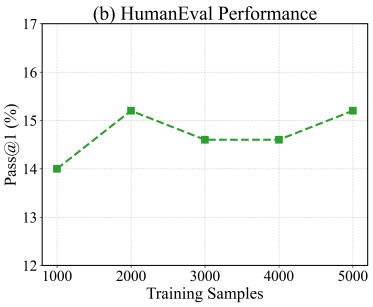

### C.3 Fusion Granularity Analysis: Block-wise vs. Channel-wise Parameter Integration

To comprehensively evaluate the impact of fusion granularity on cross-domain performance, we conduct comparative experiments between our proposed channel-wise fusion strategy and an alternative block-wise approach. The block-wise method operates at an intermediate granularity level, partitioning weight matrices into $8 \times 8$ blocks, and applying fusion decisions at the block level rather than individual parameters or channels.

**Implementation Details.** The block-wise fusion strategy divides each weight matrix $\mathbf{W} \in \mathbb{R}^{M \times N}$ into non-overlapping $8 \times 8$ blocks. For each block $\mathbf{W}_{block}^{(i,j)}$, we compute block-level differences and apply the dual-gate mechanism to determine fusion weights at the block granularity. This approach provides a middle ground between parameter-level and channel-level fusion, potentially capturing local parameter correlations while maintaining computational efficiency.

**Experimental Results.** Table 11 presents the performance comparison between single-domain models and their block-wise fused counterparts across multiple benchmarks. The results reveal distinct granularity-dependent performance patterns:

**Granularity-Performance Analysis.** The differential performance patterns suggest that fusion granularity exhibits task-dependent optimization characteristics:

- **Mathematical Reasoning Tasks:** It consistently benefits from block-wise fusion, suggesting mathematical reasoning leverages local parameter structures.
- **Code Generation Tasks:** It shows preference for channel-wise precision across all fusion combinations.
- **Multimodal Understanding:** Both approaches achieve almost equivalent scores, showing robustness to moderate granularity variations.

**Implications.** The systematic analysis reveals that fusion granularity sensitivity correlates with domain-specific computational patterns. Mathematical reasoning benefits from structured parameter blocks, while programming tasks require fine-grained control. Our channel-wise approach provides optimal average performance across diverse domain combinations, validating its adoption as the primary fusion strategy.

### C.4 Subtask-Level Further Evaluation.

Tables 12 and 13 provide comprehensive subtask-level analysis that elucidates the granular effectiveness of our fusion approach across diverse cognitive and domain-specific capabilities. The MME benchmark (Fu et al., 2023)decomposition (Table 12) reveals distinct patterns of cross-domain synergy, with Math&Medical fusion achieving superior performance in reasoning-intensive tasks (433.6 vs. 400.0 for pure Math) and code reasoning (110.0 vs. 85.0 for pure Code), demonstrating the complementary nature of mathematical and medical domain expertise in complex visual reasoning scenarios.

The subtask analysis reveals task-specific fusion benefits that align with cognitive task requirements. Visual reasoning tasks such as artwork interpretation (144.0 for Math&Code vs. 126.8 for Math

**Table 11:** Performance of single-domain and block-wise fused models across multiple domains. (✓ indicates the domain(s) included in the model).

| Domain Composition | | | | Benchmark Scores | | | |
|---|---|---|---|---|---|---|---|
| **Math** | **Code** | **Finance** | **Medical** | **MathVista** | **HumanEval** | **MMMU** | **MME** |
| ✓ | | | | 49.9 | 4.3 | 35.6 | 1455.7 |
| | ✓ | | | 47.6 | 15.9 | 37.8 | 1373.9 |
| | | ✓ | | 43.8 | 8.5 | 36.7 | 1414.7 |
| | | | ✓ | 46.0 | 12.2 | 37.5 | 1470.2 |
| ✓ | ✓ | | | 53.0 | 14.6 | 37.6 | 1486.0 |
| ✓ | | ✓ | | 50.5 | – | 37.2 | 1474.8 |
| ✓ | | | ✓ | 52.9 | – | 37.3 | 1524.3 |
| | ✓ | ✓ | | – | 14.6 | 38.1 | 1441.4 |
| | ✓ | | ✓ | – | 15.9 | 37.9 | 1470.5 |

alone) and celebrity recognition (130.3 for Math&Code vs. 113.2 for Math alone) benefit significantly from cross-domain knowledge integration, while tasks requiring specialized domain knowledge, such as OCR (125.0 for Math) and text translation (177.5 for Math), maintain optimal performance with domain-specific expertise. Notably, certain tasks demonstrate ceiling effects (existence detection achieving 195.0 across all configurations), indicating that our fusion approach preserves maximum performance capabilities without introducing degradation.

The MMMU(Yue et al., 2024) domain analysis (Table 13) confirms the effectiveness of our fusion methodology across academic disciplines. Code&Finance fusion excels in Humanities & Social Science (55.0 vs. 54.4 for Code alone) and Health & Medicine (39.6 vs. 38.2 for Finance alone), while Code&Medical fusion demonstrates consistent improvements in Science (32.5 vs. 31.9 for Code alone) and balanced performance across multiple domains. The minimal performance variations (±1.0 point maximum) across domain combinations validate the stability of our dual-gate mechanism in preventing catastrophic interference while enabling selective knowledge transfer. These fine-grained results substantiate that our parameter fusion strategy operates effectively at the cognitive subtask level, preserving specialized capabilities while enhancing cross-domain reasoning through principled parameter integration.

**Table 12:** Results of MME benchmark across tasks and domains.

| Model | math | code | math&code | medical | math&medical |
|---|---|---|---|---|---|
| reasoning | 400.0 | 357.1 | 425.0 | 372.1 | 433.6 |
| OCR | 125.0 | 57.5 | 80.0 | 95.0 | 117.5 |
| artwork | 126.8 | 115.0 | 144.0 | 138.0 | 143.25 |
| celebrity | 113.2 | 105.6 | 130.3 | 129.7 | 126.5 |
| code reasoning | 75.0 | 85.0 | 97.5 | 77.5 | 110.0 |
| color | 165.0 | 170.0 | 175.0 | 170.0 | 175.0 |
| commonsense reasoning | 105.0 | 97.1 | 105.0 | 82.1 | 108.6 |
| count | 118.3 | 115.0 | 130.0 | 120.0 | 135.0 |
| existence | 195.0 | 195.0 | 195.0 | 195.0 | 195.0 |
| landmark | 160.8 | 166.8 | 168.75 | 169.3 | 170.5 |
| numerical calculation | 42.5 | 50.0 | 52.5 | 50.0 | 52.5 |
| position | 150.0 | 128.3 | 148.3 | 143.3 | 148.3 |
| posters | 150.7 | 159.5 | 159.5 | 143.2 | 165.0 |
| scene | 151.0 | 161.25 | 157.5 | 166.8 | 159.0 |
| text translation | 177.5 | 125.0 | 170.0 | 162.5 | 162.5 |
| Overall | 1455.7 | 1373.9 | 1488.4 | 1470.2 | 1535.0 |

# D  TRAINING SETTINGS

## D.1  COMPUTATIONAL RESOURCES

All experiments were conducted on Nvidia A6000 GPUs with 48GB of RAM. Depending on the dataset type and size, as well as the fine-tuning type, for example, fine-tuning the Qwen2-VL-2B model on single tasks took between 80 minutes and 75 hours.

**Table 13:** Results of MMMU benchmark across tasks and domains.

| Model | code | finance | code&finance | medical | code&medical |
|---|---|---|---|---|---|
| Art & Design | 53.6 | 51.4 | 54.6 | 53.3 | 53.7 |
| Business | 31.4 | 30.3 | 31.4 | 29.6 | 30.5 |
| Science | 31.9 | 31.9 | 31.6 | 31.4 | 32.5 |
| Health & Medicine | 38.0 | 38.2 | 39.6 | 38.6 | 38.6 |
| Humanities & Social Science | 54.4 | 51.2 | 55.0 | 52.7 | 54.7 |
| Tech & Engineering | 34.1 | 32.3 | 33.5 | 34.6 | 34.2 |
| Overall | 37.8 | 36.7 | 38.1 | 37.5 | 38.0 |

Graft experiments took less resources. The fusion process typically completed within 1-2 hours on average, representing a considerable reduction in training time while achieving superior cross-domain performance.

## D.2 HYPER-PARAMETER SETTINGS

The training of our GraftModel method involves several key hyper-parameters and settings to ensure effective fusion of the base and graft Qwen2-VL-2B(Wang et al., 2024b) models. The specific details are summarized in Table 14. We set the learning rate to $1.0 \times 10^{-5}$ and trained for a single epoch to prevent overfitting to domain-specific patterns. The training utilized a batch size of 1 per device with gradient accumulation steps of 2, resulting in an effective batch size of 2. We implemented cosine learning rate scheduling with a warmup ratio of 0.1 to ensure smooth convergence during the fusion process. Training was conducted in bfloat16 precision with a maximum sequence length of 2048 tokens, utilizing 16 preprocessing workers for efficient data handling.

**Table 14:** Training hyper-parameters for GraftModel

| Hyper-parameter | Value |
|---|---|
| Finetuning Type | full |
| Maximum Sequence Length | 2048 tokens |
| Preprocessing Workers | 16 |
| Batch Size (per device) | 1 |
| Gradient Accumulation Steps | 2 |
| Effective Batch Size | 2 |
| Learning Rate | $1.0 \times 10^{-5}$ |
| Training Epochs | 1.0 |
| Learning Rate Schedule | cosine |
| Warmup Ratio | 0.1 |
| Precision | BF16 |

For the GraftLoRA fusion method, it targets both attention and MLP layers with LoRA adaptations, requiring a higher learning rate of $1.0 \times 10^{-4}$ and extended training over 3 epochs to achieve optimal adapter integration as shown in Table 15.

Both configurations utilize cosine learning rate scheduling with 10% warmup ratio and BF16 precision to optimize training stability and computational efficiency. The distinct hyper-parameter settings reflect the fundamental differences between full parameter fusion and adapter-based integration, with GraftLoRA requiring more aggressive optimization due to the constrained parameter space of low-rank adaptations.

To demonstrate the robustness of Graft, we further performed a comprehensive sensitivity analysis for all three parameters: entropy bin count $n$, and the arctan scaling constants $a$ and $c$.

### D.2.1 SENSITIVITY TO ENTROPY BIN COUNT($n$)

We evaluate performance across $n \in \{5, 10, 15, 20\}$.

The results in Table 16 reveal a clear trade-off between performance and efficiency:

**Table 15:** Training hyper-parameters for GraftLoRA

| Hyper-parameter | Value |
|---|---|
| LoRA Target | attention, mlp |
| Maximum Sequence Length | 2048 tokens |
| Preprocessing Workers | 16 |
| Batch Size (per device) | 4 |
| Gradient Accumulation Steps | 8 |
| Effective Batch Size | 32 |
| Learning Rate | $1.0 \times 10^{-4}$ |
| Training Epochs | 3.0 |
| Learning Rate Schedule | cosine |
| Warmup Ratio | 0.1 |
| Precision | BF16 |

Table 16: Sensitivity to entropy bin count($n$).

| Bins($n$) | MathVista | HumanEval | Time(s) |
|---|---|---|---|
| 5 | 51.8 | 14.0 | 4485.1 |
| 10 | 52.2 | 15.9 | 7299.1 |
| 15 | 52.2 | 15.9 | 8936.4 |
| 20 | 52.1 | 16.5 | 10089.4 |

Table 17: Sensitivity to arctan slope($c$).

| Slope($c$) | MathVista | HumanEval |
|---|---|---|
| 200 | 51.9 | 15.2 |
| 400 | 52.0 | 15.2 |
| 500 | 52.2 | 15.9 |
| 600 | 52.0 | 15.9 |

Table 18: Sensitivity to arctan slope($a$).

| Slope($a$) | MathVista | HumanEval |
|---|---|---|
| 0.1 | 52.2 | 13.4 |
| 0.2 | 52.2 | 15.2 |
| 0.4 | 52.2 | 15.9 |
| 0.6 | 52.2 | 15.9 |

- **Performance:** On MathVista, performance increases from $n = 5$ to $n = 10$ (from 51.8 to 52.2) and then stabilizes, peaking at $n = 10$ and $n = 15$. On HumanEval, performance steadily improves with more bins. There is a significant jump from $n = 5$ to $n = 10$ (+1.9), but performance plateaus between $n = 10$ and $n = 15$, with only a marginal gain at n=20.

- **Time Cost:** The computational time shows a strong positive correlation with $n$, increasing substantially with each increment. Moving from $n = 10$ to $n = 20$ increases the time cost by nearly 40%.

- **Trade-off Decision:** Considering these factors, $n = 10$ emerges as the optimal balance point. While a higher bin count like $n = 20$ offers a slight performance gain on HumanEval, it comes at a significant computational cost and offers no advantage on MathVista. The configuration with $n = 10$ achieves peak or near-peak performance across both benchmarks at a considerably lower time cost than larger bin counts. This choice ensures a practical balance between effectiveness and efficiency.

### D.2.2 SENSITIVITY TO ARCTAN SLOPE($c$)

We evaluate performance for $c \in \{200, 400, 500, 600\}$.

The results in Table 17 demonstrate the high stability of our method. Performance on MathVista varies by only 0.3 points across this wide range. The chosen value of $c = 500$ corresponds to the peak performance achieved on both benchmarks, making it an empirically well-justified selection.

### D.2.3 SENSITIVITY TO ARCTAN RANGE($a$)

We evaluate the impact of the arctan range parameter $a \in \{0.1, 0.2, 0.4, 0.6\}$.

As Table 18 illustrates, performance is perfectly stable on MathVista, achieving the top score of 52.2 across all tested values, and it peaks at $a = 0.4$ on HumanEval. This makes $a = 0.4$ the clear optimal choice, as it maximizes performance on both tasks.

## E   DATASET DETAILS

This section provides comprehensive information about the training datasets and evaluation benchmarks employed in our experiments. Our evaluation framework encompasses both domain-specific

fine-tuning datasets and comprehensive multimodal benchmarks to validate cross-domain fusion effectiveness.

### E.1 TRAINING DATASETS

Table 19 summarizes the domain-specific datasets used for fine-tuning specialized models. The datasets span four critical domains: mathematics, programming, finance, and medical imaging, providing diverse multimodal reasoning challenges.

**Table 19:** Detailed Description of Training Datasets

| Dataset | Domain | #Train | Modality |
|---|---|---|---|
| MathV-360K(Shi et al., 2024) | Mathematics | 338,721 | Vision+Text |
| Code-Alpaca-20K(Chaudhary, 2023) | Programming | 20,021 | Text |
| Sujet-Finance-QA-Vision-100K(Sujet AI, 2024) | Finance | 100,629 | Vision+Text |
| PathVQA(He et al., 2020) | Medical | 19,654 | Vision+Text |
| MetaMathQA-40K(Yu et al., 2023b) | Mathematics | 40,000 | Text |
| Code-Instructions-120K-Alpaca(iamtarun, 2023) | Programming | 121,959 | Text |

### E.2 DATA FORMAT

Our training datasets adopt two distinct organizational structures in order to meet different modality requirements and task characteristics.

**Multimodal Question Format.** For vision-language tasks (MathV-360K(Shi et al., 2024), Sujet-Finance-QA-Vision-100K(Sujet AI, 2024), PathVQA(He et al., 2020)), data follows a conversational structure with two primary components:

- **Message Exchange:** User-assistant dialogue pairs where user messages contain visual references ("<image>") alongside textual queries, and assistant messages provide domain-specific responses.
- **Image References:** Explicit file paths linking visual content to the corresponding textual interactions.

**Instruction-Response Format.** For text-only tasks (Code-Alpaca-20K(Chaudhary, 2023), Code-Instructions-120K-Alpaca(iamtarun, 2023), MetaMathQA-40K(Yu et al., 2023b)), data employs a structured triplet organization:

- **Instruction:** Task specifications and objectives that define the expected behavior or solution approach.
- **Input:** Additional context or parameters required for task completion (may be empty for self-contained instructions).
- **Output:** Target responses demonstrating correct task execution, including code implementations or mathematical solutions.

### E.3 EVALUATION BENCHMARKS

Table 20 details the comprehensive evaluation benchmarks used to assess both single-domain expertise and cross-domain fusion effectiveness. These benchmarks cover diverse cognitive capabilities including mathematical reasoning, code generation, multimodal understanding, and domain-specific knowledge assessment.

## F BASELINE DETAILS

This section provides a comprehensive description of the baseline methods employed in our comparative experiments. We evaluate our approach against the following established methods:

**Vector-Based Merging Methods**

**Table 20:** Detailed Description of Evaluation Benchmarks

| Benchmark | Domain | #Test | Metrics |
|-----------|--------|-------|---------|
| Math-500 (Lightman et al., 2023) | Mathematics | 500 problems | Accuracy (%) |
| MATH (Hendrycks et al., 2021b) | Mathematics | 5,000 problems | Accuracy (%) |
| MathVista (Lu et al., 2024) | Mathematics | 12 subtasks | Accuracy (%) |
| GSM8K (Cobbe et al., 2021) | Mathematics | 1,319 problems | Accuracy (%) |
| HumanEval (Chen et al., 2021) | Programming | 164 problems | Pass@1 (%) |
| HumanEval+ (Liu et al., 2023b) | Programming | 164 problems | Pass@1 (%) |
| MBPP (Austin, 2021) | Programming | 374 problems | Pass@1 (%) |
| MBPP+ (Liu et al., 2023b) | Programming | 374 problems | Pass@1 (%) |
| MMLU (Hendrycks et al., 2021a) | Knowledge | 14,042 problems | Accuracy (%) |
| C-MMLU (Li et al., 2023a) | Knowledge | 67 subtasks | Accuracy (%) |
| TruthfulQA | Knowledge | 817 problems | Accuracy (%) |
| MMMU (Yue et al., 2024) | Multimodal | 6 disciplines | Accuracy (%) |
| MME (Fu et al., 2023) | Multimodal | 14 subtasks | Composite Score |

- **Task Arithmetic:** This method introduces the concept of "task vectors" and merges these vectors into a pre-trained model to enable multi-task capability. The merged model is formulated as: $\theta_m = \theta_{init} + \lambda \cdot \sum_{t=1}^{n} \tau_t$, where $\theta_{\text{init}}$ is the initial pre-trained model, $\tau_t$ represents task-specific vectors, and $\lambda$ controls the merging intensity.

- **Ties-Merging:** This approach addresses task conflicts present in Task Arithmetic through a three-step process: Trim (eliminating redundant parameters), Elect Sign (resolving directional conflicts), and Disjoint Merge (combining non-conflicting parameters). This methodical approach enhances multi-task performance by reducing parameter interference.

**Module-Based Merging Methods**

- **DARE:** This method enhances parameter efficiency by setting the majority of delta parameters to zero and rescaling the remaining parameters: $\theta' = \theta \cdot \frac{1}{1-p}$, where $p$ represents the proportion of delta parameters eliminated. This approach effectively reduces parameter redundancy while maintaining performance.

## G  THE USE OF LLMs

In this work, we employ LLMs in two strictly controlled ways. First, LLMs are utilized to standardize the format of source data during dataset construction (see Appendix E). Second, we employ LLMs to identify and correct grammatical errors in our manuscript, ensuring clarity and linguistic accuracy throughout the paper.

## H  DISCUSSION AND OUTLOOK

### H.1  BROADER IMPACT

The Graft framework advances parameter synergy for Large Language Models (LLMs) by enabling efficient domain knowledge integration without expensive retraining. This approach significantly lowers the barrier to entry for developing powerful, multi-domain AI systems. This democratization of technology allows smaller research labs, academic institutions, and startups to create highly capable models that were previously only achievable by organizations with vast computational resources. This fosters a more inclusive and innovative AI ecosystem.

Instead of training monolithic, general-purpose models from scratch, which consumes enormous amounts of energy, the community can focus on developing and sharing smaller, efficient expert models. Graft provides the bridge to combine these experts as needed, promoting a modular, reusable, and ultimately more environmentally friendly approach to building increasingly intelligent systems.

## H.2 FUTURE DIRECTIONS

In this work, we envision several exciting avenues for future research that could further enhance capabilities of Graft and expand its applicability.

One promising direction is the development of dynamic expert selection. We aim to identify the most compatible expert models from a large repository for a given downstream task. This could evolve into a dynamic fusion mechanism where the contributions of different experts are weighted in real-time based on the specific context of an input query, creating a truly adaptive and context-aware model.

Furthermore, we plan to investigate cross-architecture grafting. While the current work focuses on fusing models with homogeneous backbones, a significant breakthrough would be to enable knowledge transfer between models of different architectures. Adapting Graft to bridge these architectural divides would unlock unprecedented opportunities for leveraging the collective knowledge of the entire AI community, regardless of the specific model family.

