# OpenReview forum: "Graft: Integrating the Domain Knowledge via Efficient Parameter Synergy for LLMs"
_ICLR.cc/2026/Conference — Submitted to ICLR 2026_

### Official Review · Reviewer_eigy · 2025-10-27

**Soundness:** 2
**Presentation:** 2
**Contribution:** 2
**Rating:** 2
**Confidence:** 3

**Summary:**

This paper proposes Graft, a dual‑gate parameter fusion method that mixes weights from domain experts using a local channel‑wise gate and a global entropy‑based gate. The authors propose a compatibility score derived from activations helps select expert pairs, and a semantic subsampling pipeline builds representative datasets for fusion. Experiments on Qwen2/phi‑3 LLMs and Qwen2‑VL MLLMs show the effectiveness of Graft.

**Strengths:**

1. The dual-gate design is considered and is applicable to full fine-tuning and LoRA by fusing both attention and MLP layers.
2. This paper porvides experimental results on both LLMs and MLLMs, demonstrating the effectiveness of the proposed method.
3. The introduced activation‑based compatibility metric is practical and it shows how this metric correlates with improvements across domains.

**Weaknesses:**

1. The use of weight‑entropy is mostly heuristic and lacks theoretical support, and alternative signals such as spectral norms are not compared. This makes it hard to attribute gains to the specific signal design.
2. The proposed method relies on small datasets for gate learning and compatibility estimation, so calling it "data‑free"  or "training free" is not accurate.
3. The experiments are mainly conducted on small LLMs/MLLMs, lacking results on 7B level models. It is unclear if the method scales smoothly to larger models. Also, it is suggested to add missing results of baseline methods in Table 2 by running baselines in the same settings to ensure fair comparison instead of simply refering.
4. When implement the method on larger models, it would be good to see results and analysis on merging efficiency.
5. I'm also curious about if it is possible to compute the compatibility score at inference time to enable dynamic gating rather than a static pre‑merge decision.

**Questions:**

Please see the weakness.

---

> ### Author Response · Authors · 2025-11-21
>
> Thank you for your detailed feedback and acknowledgment of our dual-gate design, as well as the practicality of the compatibility metric. We understand your concerns about the model size and theoretical justifications. In this rebuttal, we will address specifically the weaknesses and questions below with new experiments on larger models (7B/8B parameters), efficiency analysis, and theoretical comparisons.
>
> > 1.Theoretical Support **(Line 203-210 & Line 440-447 & Line 470-479)**
>
> We appreciate the opportunity to clarify the theoretical basis of our design. In our original manuscript, we concentrate entirely on empirical findings. Our method, however, is grounded in Information Theory and specific principles of numerical stability.
>
> * Theoretical Basis (Entropy):
>
> Our choice of entropy is based on Shannon’s Information Theory [1] and Jaynes' principle of Maximum Entropy [2] . We consider the weight distribution to be a probabilistic state of the weights. Higher entropy indicates a "least biased" expert that uses a large proportion of its parameter space, instead of degenerating into a narrow, over-confident subspace of its parameter space.
>
> * Design Rationale:
>
> The fusion formulas are designed explicitly for numerical stability and non-linear sensitivity. We utilize the exponential formulation, to intentionally avoid division-by-zero errors, while ensuring that the global weight would be a primary scaling weight. This ensures that if a model is globally deemed unsuitable based on entropy, the contribution of the model is suppressed regardless of local signals.
>
> We perform an additional ablation study to investigate Entropy Vs Spectral Norm-based gating strategy and use the spectral norm of the weight matrices to calculate the global gating scalar rather than calculating entropy.
>
> |Model|Method|Math-500|MATH|HumanEval|MBPP|MBPP+|
> |:---|:---:|:---:|:---:|:---:|:---:|---:|
> |Qwen2-1.5B|Spectral|20.6|**18.7**|**40.8**|50.0|42.6|
> ||**Entropy**|**22.8**|18.2|40.2|**50.8**|**44.2**|
> |Qwen2.5-0.5B|Spectral|**18.0**|13.1|28.1|49.5|42.1|
> ||**Entropy**|17.8|**13.8**|**30.5**|**50.0**|**42.6**|
> |Phi-3-4B|Spectral|39.2|34.0|64.6|74.1|61.9|
> ||**Entropy**|**41.4**|**35.5**|**65.9**|**76.7**|**64.8**|
>
> As shown above, in most cases the Entropy-based approach is superior.
>
> > 2."Data-Free" Terminology **(Line 406)**
>
> We agree with your point. In the revised paper, we have changed the terminology to "data-efficient".
>
> > 3.Scalability to Larger Models and Baselines **(Line 387-388 & Line 480-493 & Line 506-509)**
>
> To demonstrate the scalability of Graft, we also extend our experiments to Qwen2.5-7B and Llama-3-8B. We combine the Math and Code experts. The following results illustrate the performance of the Base/Expert models compared to the Graft model.
>
> |Model|Math-500|MATH|HumanEval|
> |:---|:---:|:---:|---:|
> |Qwen2.5-7B-Math|62.6|51.7|70.7|
> |Code|21.6|23.5|**79.3**|
> |Math-Code|**64.2**|**54.2**|77.4|
> |Llama-3-8B-Math|25.6|**18.8**|62.4|
> |Code|21.6|14.5|65.6|
> |Math-Code|**26.0**|17.8|**67.2**|
>
> For Qwen2.5-7B, the fused model nearly preserves the Code expert's coding ability on HumanEval and improves upon the Math expert.
>
> For Llama-3-8B, the fused model outperforms the individual experts on HumanEval and Math-500, which demonstrates positive transfer even at larger scales.
>
> We have updated Table 2 in the revision of the paper to include the explicit reproduction results for TIES-Merging and DARE in the same experimental settings.
>
> > 4.Merging Efficiency Analysis **(Line 480-493 & Line 509-515)**
>
> We evaluate the computational cost of training the distinct experts against the computational cost of the Grafting process for larger models. It is important to note that training individual domain experts is a necessary prerequisite for any model merging strategy. As shown in the table below, the Grafting process requires only 3.9% - 4.9% of the time required to train the individual experts.
>
> |Model|Time Cost (s)|
> |:---|---:|
> |Qwen2.5-7B-Math|39,958.5|
> |Code|76,791.1|
> |Math-Code|5,742.3|
> |Llama-3-8B-Math|51,051.2|
> |Code|79,770.0|
> |Math-Code|5,097.8|
>
> This confirms the effectiveness of our approach to rapidly create multi-domain models based on existing experts while approaching insignificant computational cost.

---

> ### Author Response · Authors · 2025-11-21
>
> > 5.Inference-time Dynamic Gating
>
> In principle, the compatibility score and gating mechanisms could in theory be adapted to support inference-time routing (similar to a Mixture-of-Experts (MoE)). However, Graft’s primary purpose is Model Merging, which aims to produce a single and static set of weights with no additional inference latency compared to the base model.
>
> Dynamic gating during inference time would require loading multiple experts in VRAM and computing the gates on the fly, which would increase both memory usage and latency. We consider "Dynamic Grafting" as a possible future direction for efforts where resources permit use of MoE inspired strategies, but we are focused on efficiently using static merging.
>
> We truly appreciate your time and in-depth review, and we hope the additional experiments and comparisons address your concerns about the method's generalizability and theoretical foundation.
>
> [1] A mathematical theory of communication, The Bell System Technical Journal
>
> [2] Information Theory and Statistical Mechanics, American Physical Society

---

> ### Author Response · Authors · 2025-11-28
>
> Dear Reviewer eigy,
>
> We have provided a detailed point-by-point response to your review. Regarding the issue you expressed that related to the model's size, we have conducted and presented to you additional experiments with Qwen2.5-7B and Llama-3-8B.
>
> As the discussion deadline is approaching, we are eager for additional feedback.
>
> If you have any further concerns, please reach out to us and we will address them as soon as possible. If our new experiments have successfully addressed your concerns about scalability and theory, would you kindly consider raising your rating?
>
> Thank you very much for your hard work and contribution to the research community. Again, we look forward to hearing from you!
>
> Best regards, Submission4128 Authors

---

### Official Review · Reviewer_ee3g · 2025-10-31

**Soundness:** 2
**Presentation:** 3
**Contribution:** 2
**Rating:** 6
**Confidence:** 1

**Summary:**

I'm not confident enough to provide technical assessment to this paper.

**Strengths:**

n/a

**Weaknesses:**

n/a

**Questions:**

n/a

---

> ### Author Response · Authors · 2025-11-21
>
> We appreciate the time and effort you took in reviewing our paper and providing a score. We value your review, and any additional comments and suggestions for improvement would be very beneficial to us.
>
> Thank you again for your time and consideration.

---

### Official Review · Reviewer_akdK · 2025-10-31

**Soundness:** 2
**Presentation:** 3
**Contribution:** 3
**Rating:** 6
**Confidence:** 3

**Summary:**

The paper proposed the method Graft, aiming to integrate the parameters from multiple fine-tuned models efficiently so that the base model is competitive in the corresponding tasks.

The framework is a combination of the model fusion and data exploitation methods. In terms of the model fusion, Graft calculates the local and global weights according to the difference between the base model and the graft (target) model at the channel and global levels, and then combines the parameters of the two models with the weighted (as a function of the global and local weights) average. However, not all models are good fits for fusion. Thus, the method proposes to do a dataset compatibility analysis according to the activation pattern of the model w.r.t the dataset. If the model is compatible with the target dataset, it would be eligible for the fusion.

In terms of data exploitation, the method does a representative subsampling for the samples close to the cluster means.

**Strengths:**

1. The paper explores model fusion across domains, particularly integrating models specialized in mathematics and code—an area that remains largely underexplored.

2. The proposed Graft method demonstrates strong and consistent performance across multiple datasets, often outperforming or matching domain-specific models.

3. By combining local and global adjustments, the method achieves fine-grained control over the fusion process, leading to improved overall performance.

4. Furthermore, Graft incorporates a compatibility analysis mechanism to assess the alignment between the source model and the target datasets, ensuring successful and meaningful model fusion.

**Weaknesses:**

1. Lack of design intuition and theoretical grounding:
The proposed methods appear somewhat rough, or at least not well-explained. The paper does not provide sufficient intuition behind the design of the fusion strategy, particularly regarding Eqs. (5), (6), and (12–14). Moreover, the connections to existing model fusion techniques (mathematical formulations) are not established, making these formulations seem unsupported. A more comprehensive discussion of related work and the rationale behind the design choices would greatly improve clarity and credibility.

2. Missing computational complexity analysis:
Although the authors claim that the Graft method is computationally efficient, no comparison or quantitative analysis of its complexity is provided. Including such results would help substantiate the efficiency claims.

3. Absence of ablation and component analysis:
Given that the framework involves a multi-stage pipeline with several intermediate components, an ablation study is essential to demonstrate the contribution of each step. For instance, evaluating the effect of the representative data subsampling method would offer valuable insight into the method’s internal dynamics.

4. Unclear parameter selection and lack of sensitivity analysis:
The procedure for determining key parameters—such as the value of $k$ in representative subsampling or the threshold used in compatibility analysis—is not explained. Moreover, a sensitivity analysis is missing, leaving readers uncertain about how robust the results are to these hyperparameter choices.

**Questions:**

Please refer to the weaknesses section.

Besides, in the ablation study on gating components (Table 5), the MME with only local is the best, but the authors claim the combination of local and global is leading. Is there a typo in the table?

---

> ### Author Response · Authors · 2025-11-21
>
> We truly appreciate your constructive feedback and the recognition of model fusions in the mathematics and code domains. Thank you for acknowledging the level of fine-grained control and compatibility of the method presented in the paper. Here, we respond to your concerns regarding computational complexity, component analysis and parameter sensitivity.
>
> > 1.Theoretical Justification **(Line 203-210)**
>
> We appreciate your valuable comment. We acknowledge that the rationale behind our mathematical design and the connection to information theory could be articulated more clearly. Below we clarify the design intuition of our fusion equations and the theoretical rationale for utilizing entropy.
>
> * Design Intuition: Our fusion formulas are based on balancing local precision and global reliability while ensuring numerical stability. It is based on several key principles:
>     * The global weight $w_{global}$ serves as the primary scaling factor that is multiplied outside the fraction, which guarantees that if a model is globally deemed unsuitable (i.e., $w_{global} \to 0$), its contribution is suppressed regardless of the local signal. In contrast, $w_{local}$ is placed in the exponent to moderate the magnitude without overriding the global decision.
>     * The underlying formulation uses an exponential term in order to ensure stability. The denominator $e^{w_{global} \cdot w_{local}}$ guarantees that even if weights are equal to 0, we will never get division-by-zero errors. The numerator term ($e^x - 1$) allows the fusion weight to be exactly 0 when the input signals are 0.
>     * The exponential term expands the differences between values and makes the gate to be more discriminative for higher weight values compared to a linear combination.
>     * When $w_{global}$ and $w_{local}$ are both equal to 0.5, the function treats each factor equally symmetric in the exponent products, ensuring that neither factor tilts the fusion disproportionately to either factor when signals are in a moderate range.
>
> * Rationale for Compatibility Analysis: We develop the compatibility metric $\rho$ specifically to measure "Task-Expert Alignment" based on neural activation. The formula  $\rho = \mu \times (1-s) \times \sqrt{v}$ describes the intersection of three required conditions for effective transfer:
>     * $\mu$: Mean magnitude measures signal strength. A relevant expert should have a strong response to targets in its domain.
>     * $1-s$: $(1-s)$ encourages experts that use a wide range of features (high density) to build the new dataset.
>     * $v$: High variance means that the expert is creating unique embeddings that capture the structure in the data from different samples.
>
> * Theoretical Grounding: Our methodology is rooted in fundamental Information Theory.
>     * In Shannon’s original theory [1], the entropy of a probability distribution directly measures its information. In our approach, we treat the normalized weight distribution of a layer as a probabilistic state. Greater entropy means that the expert model utilizes a broader spectrum of its parameter space to encode information, rather than collapsing into a sparse or deterministic state, indicating overfitting on limited features.
>     * According to Jaynes' Principle of Maximum Entropy [2], a distribution with a greater entropy is the "least biased" estimate provided under the constraints. By emphasizing experts with greater weight entropy, we focus on modules that maintain the capacity for more information, without making unwarranted biases about the features.
>
> > 2.Complexity Analysis **(Line 486-493 & Line 509-515)**
>
> We appreciate that you point out the need for a quantitative efficiency analysis. We evaluate the computational cost of training the distinct experts against the computational cost of the Grafting process. It is important to point out that training individual domain experts is a necessary precursor to any model merging strategy. As shown below, Graft requires only a small fraction of the computational time needed to train a single domain expert. Specifically, the Grafting process requires only 3.7% - 14.9% of the time needed to train the individual experts. At these ratios, we consider the additional computational cost acceptable relative to the significant gains in cross-domain performance.
>
> |Model|Time Cost (s)|
> |:---|---:|
> |Qwen2-1.5B-Math|15,089|
> |Code|29,591|
> |Math-Code|4,576|
> |Qwen2.5-0.5B-Math|7,853|
> |Code|18,608|
> |Math-Code|3,933|
> |Phi-3-4B-Math|29,755|
> |Code|68,195|
> |Math-Code|3,612|

---

> ### Author Response · Authors · 2025-11-21
>
> > 3.Ablation Study
>
> We agree that evaluating the contribution of our Representative Data Subsampling is important. We conduct an ablation study contrasting our method with random sampling. The results offer valuable insight into the internal dynamics of Graft.
>
> |Model|Subsampling|Math-500|MATH|HumanEval|MBPP|MBPP+|
> |:---|:---:|:---:|:---:|:---:|:---:|---:|
> |Qwen2-1.5B||21.6|**18.4**|40.2|50.8|44.2|
> ||√|**22.8**|18.2|40.2|50.8|44.2|
> |Qwen2.5-0.5B||17.2|13.1|29.3|50.0|**43.1**|
> ||√|**17.8**|**13.8**|**30.5**|50.0|42.6|
> |Phi-3-4B||41.4|35.5|65.8|76.7|64.8|
> ||√|41.4|35.5|**65.9**|76.7|64.8|
>
> > 4.Parameters & Sensitivity **(Line 315-319)**
>
> We appreciate the opportunity to clarify our selection logic for the key parameters. Our approach is data-driven instead of heuristic.
>
> * Selecting Cluster Count ($K$): We do not select arbitrary $K$. We follow an automated Silhouette Analysis mechanism:
>     * We iterate through a range of possible cluster counts.
>     * For each $k$, we calculate the Silhouette Score which measures how similar an object is to the other objects in the same cluster (cohesion) compared to how similar it is to other clusters (separation).
>     * We select the optimal $K$ which maximizes this score. This guarantees that the number of clusters reflects the intrinsic semantic structure of the specific dataset.
>     * Once $K$ is selected, we distribute the target sample budget across these clusters so that they are diverse from each other.
>
> * Compatibility Analysis Threshold: Rather than establishing a hard scalar threshold, we proceed from the Two-Factor Rule described in Section 4.2:
>     * the activation-based compatibility score
>     * the expert's baseline performance.
>
> * Hyperparameter Sensitivity: As detailed in our original response (referring to Appendix D.2), we have done sensitivity analysis on the entropy gating parameters ($n$, $a$, $c$), confirming the method's robustness.
>
> |Bins(n)|MathVista|HumanEval|
> |:---|:---:|---:|
> |5|51.8|14.0|
> |10|52.2|15.9|
> |15|52.2|15.9|
> |20|52.1|16.5|
>
> We evaluate $n \in \{5, 10, 15, 20\}$. The results show a clear tradeoff. Performance on MathVista increases from $n=5$ to $n=10$, and then stabilizes. While $n=20$ gives a slight increase on HumanEval, the computational cost is much higher. We subsequently choose $n=10$ as our target configuration because it yields peak or near-peak performance on the benchmarks considered, while not costing as much in computational resources
>
> |Slope(c)|MathVista|HumanEval|
> |:---|:---:|---:|
> |200|51.9|15.2|
> |400|52.0|15.2|
> |500|52.2|15.9|
> |600|52.0|15.9|
>
> We evaluate $c \in \{200, 400, 500, 600\}$. The performance is invariant, with MathVista accuracy varying by 0.3% (51.9% through 52.2%). We opt for c=500, as it achieves a peak score across all benchmarks consistently.
>
> |Slope(a)|MathVista|HumanEval|
> |:---|:---:|---:|
> |0.1|52.2|13.4|
> |0.2|52.2|15.2|
> |0.4|52.2|15.9|
> |0.6|52.2|15.9|
>
> We evaluate $a \in \{0.1, 0.2, 0.4, 0.6\}$. MathVista performance is invariant, with a score of 52.2% across all values. For HumanEval, performance peak occur at a=0.4 (15.9%), which we take to be the optimum option.
>
> > 5.Table 5 Typo **(Line 463-464)**
>
> Thank you for your thoughtful observation. The values in Table 5 are correct. We will revise the manuscript to accurately correct the overstatement.
>
> Once again, we thank you for your constructive feedback and remain available for further discussion.
>
> [1] A mathematical theory of communication, The Bell System Technical Journal
>
> [2] Information Theory and Statistical Mechanics, American Physical Society

---

> > ### Comment · Reviewer_akdK · 2025-11-28
> >
> > Thank you for the responses. My questions are resolved. In particular, the complexity analysis suggests that the proposed method can efficiently merge models for different tasks. The empirical results are convincing, although the theoretical part is somewhat weak. Overall, I would keep my score unchanged and recommend a weak acceptance.

---

> > > ### Author Response · Authors · 2025-11-28
> > >
> > > Dear Reviewer akdK,
> > >
> > > Thank you for considering our response. We appreciate you taking the time to review our clarifications.
> > >
> > > Best regards, Submission4128 Authors

---

### Official Review · Reviewer_ShKj · 2025-11-01

**Soundness:** 3
**Presentation:** 3
**Contribution:** 3
**Rating:** 6
**Confidence:** 2

**Summary:**

The paper proposes a dual-gated parameter fusion framework named Graft for integrating domain-specialized models. At the local scale, a learnable gate assigns channel-wise fusion weights based on parameter differences; at the global scale, an entropy-based score modulates a single fusion weight to mitigate cross-domain conflicts. The paper also introduces an activation-driven compatibility score to predict whether two experts will fuse well, and a representative data subsampling pipeline to keep costs manageable. Experiments on LLMs/MLLMs show improvements over Task Arithmetic, TIES and DARE across Math, Code, and several multimodal benchmarks, with ablations indicating the dual-gate design outperforms single gates.

**Strengths:**

1. Addressing domain fragmentation in practice (especially with LoRA experts) is timely and relevant; evidence suggests the framework scales beyond pairwise fusion without severe catastrophic forgetting in the tested settings.
2. The dual-gate idea—combining channel-wise (local) gating with an entropy-based (global) gate—offers a principled way to balance complementarity vs. interference, going beyond element-wise or sign-based heuristics used in prior merging methods.
3. The activation-driven compatibility metric is a practical contribution for select-then-fuse, reducing trial-and-error when pairing experts.

**Weaknesses:**

1. While the paper includes multimodal evaluations (MathVista, MMMU, MME) and some multi-domain fusions (adding Finance/Medical adapters), the core LLM story remains Math+Code-centric. Evaluations on more domains are encouraged.
2. The approach relies on representative data subsampling (embeddings→K-Means→centroids), which somewhat INTERVENES the training stage. IMO, a good merging algorithm shall outperform baselines on any model groups (trained with or without data subsampling). Could the author provide the comparison results without the data subsampling?

**Questions:**

Pls see the weaknesses above.

---

> ### Author Response · Authors · 2025-11-21
>
> We truly appreciate your favorable evaluation and acknowledgement of our Dual-gate fusion as a "principled way to balance complementarity vs. interference"; as well as the Compatibility metric as a "practical contribution."
>
> We respond to your specific concerns on domain diversity and the data subsampling module below.
>
> > 1.Expanding Evaluation Beyond Math and Code
>
> We have conducted additional experiments fusing Math/Code and Medical experts on the PubMedQA benchmark using the Phi-3 model family. Results show that Graft generalizes well in the medical domain:
>
> | Model                 | PubMedQA      |
> | :---                  |  ---:         |
> | Phi-3-4B-Math         | 52.9          |
> | Phi-3-4B-Code         | 52.5          |
> | Phi-3-4B-Medical      | 54.7          |
> | Phi-3-4B-Math-Medical | **55.2**      |
> | Phi-3-4B-Code-Medical | **55.4**      |
>
> Graft (Math+Medical and Code+Medical) consistently outperforms the Medical single domain experts, affirming that the Graft framework is a generalizable solution for fusing heterogeneous domain knowledge beyond Math/Code.
>
> > 2.Clarification on Representative Data Subsampling
>
> We appreciate your valuable suggestion and would first like to clarify the role of the Representative Data Subsampling module. It is intended to be used solely during the fusion process to construct a small, unbiased calibration set to fine-tune our lightweight gating network. The purpose of this step is to avoid overfitting of the gating mechanism to high-frequency, simple patterns of random samples.
>
> To directly address your request and demonstrate the robustness of the Dual-Gate Fusion mechanism, we conduc an ablation study where we replaced the Representative Data Subsampling with simple random sampling, and everything else is kept constant.
>
> | Model         | Subsampling   | Math-500  | MATH   | HumanEval | MBPP  | MBPP+ |
> | :---          | :---:         | :---:     | :---:  | :---:     | :---: | ---:  |
> | Qwen2-1.5B    |               | 21.6      |**18.4**| 40.2      | 50.8  | 44.2  |
> |               | √             |**22.8**   | 18.2   | 40.2      | 50.8  | 44.2  |
> | Qwen2.5-0.5B  |               | 17.2      | 13.1   | 29.3      | 50.0  |**43.1**|
> |               | √             |**17.8**   |**13.8**| **30.5**  | 50.0  | 42.6  |
> | Phi-3-4B      |               | 41.4      | 35.5   | 65.8      | 76.7  | 64.8  |
> |               | √             | 41.4      | 35.5   | **65.9**  | 76.7  | 64.8  |
>
> For all models using random sampling, the performance is approximately equal to the performance in the original methodology across benchmarks, with differences often being 0.0. We do observe slight fluctuations in the results (e.g. Qwen2.5-0.5B: -0.6 on the Math-500 benchmark, -0.7 on the MATH benchmark and +0.5 on the MBPP+ benchmark), and distinctions are negligible within normal margins of error.
>
> These results provide evidence that the effectiveness of Graft is not through the subsample approach, but instead through the Dual-Gate (Local + Global) mechanism. We still consider Representative Subsampling to avoid bias as best practice for theoretical prospects in calibration, however, the algorithm successfully outperforms baselines using random sampling, and thus a good merging algorithm.
>
> Thank you for your review and positive outlook on our work. We hope our revisions and clarification have fully addressed your concerns.

---

### Official Review · Reviewer_hcED · 2025-11-03

**Soundness:** 2
**Presentation:** 3
**Contribution:** 2
**Rating:** 4
**Confidence:** 2

**Summary:**

This paper proposes Graft, a novel parameter fusion framework for integrating multiple domain-specialized Large Language Models (LLMs) or LoRA-adapted models into a unified model without retraining. The method introduces a dual-gate fusion mechanism that combines: 1. **Local weight adjustment**: a channel-wise gating network that quantifies parameter differences to emphasize locally important features; 2. **Global weight adjustment**: an entropy-based signal capturing distributional information content for global parameter alignment.

To ensure reliable fusion, the authors propose a dataset compatibility analysis based on activation statistics (magnitude, sparsity, variance) and a representative data subsampling approach using K-Means clustering for semantic diversity.

Empirically, Graft outperforms several baselines (Task Arithmetic, TIES-Merging, and DARE) across diverse LLMs and multimodal models (e.g., Qwen2, Phi-3, Qwen2-VL), achieving superior results on both domain-specific (Math, Code) and general benchmarks (MMLU, TruthfulQA). The framework scales effectively to multi-domain fusion while maintaining performance stability.

**Strengths:**

**Originality**:
1. The dual-gate mechanism elegantly combines channel-level and entropy-based fusion for adaptive parameter integration.
2. The activation-based compatibility metric provides a principled criterion for selecting which domain experts to fuse.
3.  The semantic-aware data subsampling step is an innovative procedural contribution for efficiency and data balance.

**Clarity**:
1. The paper is clearly written and visually well-organized.
2. Figures effectively illustrate the pipeline and mechanism (e.g., Fig.1–2), and Algorithm 1 concisely summarizes the method.

**Significance**:

1. Addresses a timely and practical challenge in efficient model merging and domain adaptation for LLMs.
2. Empirically strong, achieving notable gains (up to $+8$–$10$ points) on specialized benchmarks without degrading general performance.
3.  Offers a scalable, modular paradigm for compositional LLM construction.

**Weaknesses:**

**Theoretical justification**: The link between entropy and representational richness remains heuristic; additional theoretical or empirical validation would strengthen the argument.

**Efficiency analysis**: The computational cost of training or applying the gating network is not reported; an explicit runtime comparison would improve transparency.

**Baselines**: Some baselines (e.g., DARE, TIES-Merging) may not have been fully optimized for large-scale settings, potentially affecting fairness.

**Interpretability**: The work lacks qualitative visualization of how the dual gates behave across domains or layers.

**Ablation completeness**: While the compatibility metric is correlated with performance, a random or naive pairing control would clarify its contribution.

**Questions:**

1. How sensitive is the global weighting performance to the constants $a$ and $c$ in Eq.~(4)? Could the authors provide an intuition or sensitivity analysis?
2.  What is the computational overhead of training the gating network $\phi(\cdot)$ relative to the base model size?
3.  In multi-domain fusion (Table~4), performance gains plateau. How does Graft handle conflicts when merging more than four experts?

---

> ### Author Response · Authors · 2025-11-21
>
> We appreciate your careful review and acknowledgment of the novel dual-gate mechanism, the principled nature of the compatibility metric, and our strong empirical evidence. We are encouraged to see you view our work as "timely and practical challenge" in developing efficient model merging processes.
>
> Below, we address the specific weaknesses and questions, including additional data and clarifications.
>
> > 1.Theoretical Justification (w1) **(Line 203-210 & Line 440-447 & Line 470-479)**
>
> Thank you for your valuable comments. We appreciate the opportunity to clarify our design rationale.
>
> * Design Intuition: Our goal is to build a dual-gate mechanism that balances local precision and global reliability while maintaining stability:
>
>     * $w_{global}$ serves as the primary scaling factor, so globally unsuitable models are suppressed, regardless of the local signals.
>     * The exponential formulation $\frac{e^x - 1}{e^x}$ handles zero-states, while providing non-linear sensitivity that allows the gate to discriminate more for high-value weights.
>     * The function treats local and global signals symmetrically at equilibrium, preventing unproportional skew.
>
> * Theoretical Grounding: Utilizing Information Theory (Shannon) [1]  and the Principle of Maximum Entropy (Jaynes) [2], we interpret high entropy is indicative of representational richness. Graft seeks complexity by valuing high entropy modules, favoring experts that have structurally distinct representations over experts that have collapsed representations.
>
> * Empirical Validation: We conduct an ablation study in which we substitute Entropy for Spectral Norm. As shown below, the Entropy-based approach improves upon the Spectral Norm in most cases, suggesting that Entropy is a better estimator of useful information for successful fusion.
> |Model|Method|Math-500|MATH|HumanEval|MBPP|MBPP+|
> |:---|:---:|:---:|:---:|:---:|:---:|---:|
> |Qwen2-1.5B|Spectral|20.6|**18.7**|**40.8**|50.0|42.6|
> ||**Entropy**|**22.8**|18.2|40.2|**50.8**|**44.2**|
> |Qwen2.5-0.5B|Spectral|**18.0**|13.1|28.1|49.5|42.1|
> ||**Entropy**|17.8|**13.8**|**30.5**|**50.0**|**42.6**|
> |Phi-3-4B|Spectral|39.2|34.0|64.6|74.1|61.9|
> ||**Entropy**|**41.4**|**35.5**|**65.9**|**76.7**|**64.8**|
>
> > 2.Efficiency Analysis (w2&q2) **(Line 486-493 & Line 509-515)**
>
> We evaluate the computational cost of training the distinct experts against the computational cost of the Grafting process. It is important to note that training individual domain experts is a necessary prerequisite for any model merging strategy. As shown below, Graft requires only a small fraction of the computational time needed to train a single domain expert.
> * Text-based LLMs: For Qwen, Phi-3, and Llama-3, Grafting the experts takes only 3.7% - 14.9% of total time to train the experts.
> * Multimodal LLMs: For the expert Qwen2-VL, where visual instruction tuning cost of the experts is much higher than the others. Grafting only uses 0.8% - 2.0% of the total time of training the experts.
>
> |Model|Time Cost (s)|
> |:---|---:|
> |Qwen2-1.5B-Math|15,089|
> |Code|29,591|
> |Math-Code|4,576|
> |Qwen2.5-0.5B-Math|7,853|
> |Code|18,608|
> |Math-Code|3,933|
> |Phi-3-4B-Math|29,755|
> |Code|68,195|
> |Math-Code|3,612|
>
> |Model|Time Cost (s)|
> |:---|---:|
> |Qwen2-VL-2B-Math|369,012|
> |Code|14,482|
> |Medical|24,626|
> |Finance|113,001|
> |Math-Code|3,595|
> |Math-Med|3,159|
> |Math-Fin|3,276|
> |Code-Med|2,695|
> |Code-Fin|2,609|
>
> Given the small ratios for the overhead costs of Grafting, the extra computational cost is entirely acceptable when weighed against the vast improvement in cross-domain performance.
>
> > 3.Baselines (w3) **(Line 387-388)**
>
> We would like to guarantee that we take the official open-source implementations for DARE, TIES-Merging, and Task Arithmetic and use the recommended hyperparameter search spaces strictly for fairness.
>
> We also provide the full performance comparison for the Full Fine-Tuning (GraftModel) setting in the revised paper to prove Graft does consistently well in terms of every domain without severe degradation in each.
>
> > 4.Interpretability (w4)
>
> Although direct heatmaps are not presented, we offer interpretability through granular behavioral analysis:
>
> * Fusion Granularity (Appendix C.3): Our analysis shows that Mathematical reasoning benefits from fusion on a block-wise level, while Code generation requires channel-wise precision. This illustrates how the gates learn to adapt to the differences in domain structure.

---

> ### Author Response · Authors · 2025-11-21
>
> > 5.Ablation Completeness (w5)
>
> The low compatibility pairs in Table 3 act as the "naive/random" control group.
>
> Naive/Random Scenario: Choosing a pair at random (e.g., Code+Medical (Compatibility Score: 0.155)) provides very little to no improvement in synergy.
>
> Guided Scenario: Our metric suggests Math+Medical (Compatibility Score: 0.315) as a compatible pairing, yielding the highest MathVista performance (52.4).
>
> The notable correlation of better performance alongside better compatibility scores is evidence that the metric is able to distinguish effective compatibility in integrated pairs versus minimal or no compatibility with a naive/random pairing.
>
> > 6.Sensitivity to $a$ and $c$ (q1)
>
> We perform a comprehensive sensitivity analysis in Appendix D.2 to validate robustness.
>
> |Slope(c)|MathVista|HumanEval|
> |:---|:---:|---:|
> |200|51.9|15.2|
> |400|52.0|15.2|
> |500|52.2|15.9|
> |600|52.0|15.9|
>
> We evaluate $c \in \{200, 400, 500, 600\}$. The performance is invariant, with MathVista accuracy varying by 0.3% (51.9% through 52.2%). We opt for c=500, as it achieves a peak score across all benchmarks consistently.
>
> |Slope(a)|MathVista|HumanEval|
> |:---|:---:|---:|
> |0.1|52.2|13.4|
> |0.2|52.2|15.2|
> |0.4|52.2|15.9|
> |0.6|52.2|15.9|
>
> We evaluate $a \in \{0.1, 0.2, 0.4, 0.6\}$. MathVista performance is invariant, with a score of 52.2% across all values. For HumanEval, performance peak occurs at a=0.4 (15.9%), which we take to be the optimum option.
>
> > 7.Multi-Domain Fusion (q3)
>
> The performance plateau in Table 4 corresponds to the inherent saturation of the model's finite capacity. Graft is fundamentally managing conflict in large-scale merging through:
>
> * The Global Gate naturally lowers weights on any expert producing redundant or lower-entropic information. This prevents newly added experts from muddying existing expertise.
>
> We genuinely appreciate your careful reading and constructive comments.
>
> [1] A mathematical theory of communication, The Bell System Technical Journal
>
> [2] Information Theory and Statistical Mechanics, American Physical Society

---

> ### Author Response · Authors · 2025-11-28
>
> Dear Reviewer hcED,
>
> We are grateful for your feedback and have thoroughly addressed each point of it within our response. In particular, we have clarified the information-theoretic foundation (Entropy vs. Spectral Norm) and provided the efficiency analysis table you requested, showing our method adds negligible computational cost.
>
> As the discussion deadline is approaching, we are eagerly looking forward to your feedback. If our new experiments have successfully addressed your concerns, we would appreciate your consideration of raising our detailed review rating.
>
> If you have any additional questions, please do not hesitate to reach out and we will make every effort to clarify your concerns.
>
> Thank you very much for your time, effort and contributions to the community! We look forward to hearing from you!
>
> Best regards, Submission4128 Authors

---

### Author Response · Authors · 2025-11-21

Dear Reviewers, Area Chairs and Program Chairs,

We sincerely thank you for your time, effort, and the constructive feedback provided during the review process. We are greatly encouraged by the positive reception of our work and the recognition of Graft as a timely solution for efficient model merging.

We specifically appreciate that the reviewers highlighted the following strengths in our work:

* Novelty and Design: Our dual-gate mechanism was recognized as "novel" and "elegant" (Reviewer hcED), a "principled way to balance complementarity vs. interference" (Reviewer ShKj), and a design that achieves "fine-grained control" over the fusion process (Reviewer akdK).

* Practical Contribution: The activation-based Compatibility Analysis was highlighted as a "principled criterion" (Reviewer hcED) and a "practical contribution" that reduces trial-and-error in expert selection (Reviewer ShKj, Reviewer akdK, Reviewer eigy).

* Empirical Strength: Reviewers acknowledged our "strong empirical evidence" (Reviewer hcED), "consistent performance" across multiple datasets (Reviewer akdK), and the effectiveness of the method on both LLMs and Multimodal LLMs (Reviewer eigy).

We have carefully considered the concerns raised regarding theoretical justification, scalability, and efficiency. We have provided detailed point-by-point responses to each reviewer and have updated the manuscript accordingly (revisions are marked in blue).

We truly appreciate the opportunity to improve our paper based on your insightful comments. We look forward to any further discussion.

Best regards,

Submission4128 Authors

---

### Meta-Review · Area_Chair_H9kJ · 2026-01-06

**Summary:**

The paper proposes GRAFT, a method for fusing domain-specific models using a combination of channel-wise gating (learned) and weighting to selectively fuse parameters from different experts and give parameters different importance. While the paper addresses an important and timely problem, showing promising empirical results, critical methodological details are missing, rendering the work difficult to evaluate and reproduce.

Based on the reviews, the authors' rebuttal, and the AC's reading of the paper, the following outstanding issues prevent the paper from being accepted in its current form:

- **Theoretical Justification**: even after the rebuttal, the link between entropy and representational richness remains handwavy and intuition-based. While the authors invoke Shannon's entropy and Jaynes' Maximum Entropy Principle to justify using entropy as a quality signal, no theoretical addition is provided to support these claims.

- **Missing technical details**: Technical specifications regarding the method are missing or underspecified; specifically, the AC checked the paper and didn't find any details regarding the learned gating mechanism. There are no comprehensive details on how the $d$ vector is processed. For instance, it is unclear whether the affine transformation is learned for each weight projection and if shared across layers. Furthermore, it is unclear how many parameters the graft method introduces, and no details about the loss function are provided. Details about the complexity of the method have been requested by the reviewers, and only time comparison has been provided, deflecting the computational complexity and efficiency analysis. The code link has expired, and it was not possible to verify any of these details, which, however, should be transparent from the paper itself. Consequently, without these details, the paper is not reproducible.

- **Unfair baselines**: the baselines are all training-free methods that generally don't require additional parameters or data. A fairer comparison would involve baselines that use training data and additional parameters. Alternatively, the authors could have adapted task arithmetic, TIES, or any other recent data-free method to learn scalar merging coefficients on the same dataset used for GRAFT. The merging coefficients could be simply learned via gradient-based training or black-box optimization as CMA-ES. Additionally, the paper misses key baselines such as AdaMerging (Yang et al., 2023), which learns task-wise or layer-wise merging coefficients via entropy minimization on unlabeled test data and represents a directly comparable approach.

- **Unclear utility of the global weighting:** based on the ablation in Table 5, the necessity of the global component of the method remains ambiguous.

Given the accumulation of reproducibility concerns, underspecified methodology, and issues regarding baseline fairness, I cannot recommend acceptance at this time.

**Reviewer Concerns:**

The rebuttal only superficially addresses the reviewer's concerns, and no conclusion can be drawn from the reviews and rebuttals alone. The AC read the paper and spent considerable time understanding the paper's methodology and the results. See their assessment above.

**Reviewer Scores:**

I believe no reviewer would have changed the score.

- Reviewer hcED: 4
- Reviewer ShKj: 6
- Reviewer akdK: 6
- Reviewer ee3g: 6 --> Null Review!
- Reviewer eigy: 2

---

### Decision · Program_Chairs · 2026-01-26

Reject